# Healing The Past By Nurturing The Future: A qualitative systematic review and meta-synthesis of pregnancy, birth and early postpartum experiences and views of parents with a history of childhood maltreatment

Catherine Chamberlain[1,2,3]*, Naomi Ralph[1], Stacey Hokke[1], Yvonne Clark[1,4], Graham Gee[2,5], Claire Stansfield[6], Katy Sutcliffe[6], Stephanie J. Brown[2,4,7], Sue Brennan[3], for the Healing The Past By Nurturing The Future group[¶]

1 Judith Lumley Centre, La Trobe University, Melbourne, Victoria, Australia, 2 Intergenerational Health Group, Murdoch Children's Research Institute, Melbourne, Victoria, Australia, 3 School of Public Health and Preventive Medicine, Monash University, Melbourne, Victoria, Australia, 4 Women and Kids Theme, South Australian Health and Medical Research Institute, Adelaide, South Australia, Australia, 5 Melbourne School of Psychological Sciences, University of Melbourne, Melbourne, Victoria, Australia, 6 Evidence for Policy and Practice Co-ordinating Centre, Social Science Research Unit, UCL Institute of Education, University College London, London, United Kingdom, 7 Department of Paediatrics, Royal Children's Hospital, The University of Melbourne, Melbourne, Victoria, Australia

¶ Membership of the Healing The Past By Nurturing The Future group is provided in the Acknowledgments.
* c.chamberlain@latrobe.edu.au

## Abstract

### Background

Child maltreatment can have serious effects on development and physical, social and emotional wellbeing. Any long-lasting relational effects can impede the capacity to nurture children, potentially leading to 'intergenerational trauma'. Conversely, the transition to parenthood during pregnancy, birth and the early postpartum period offers a unique life-course opportunity for healing. This systematic review aims to understand the pregnancy, birth and early postpartum experiences of parents who reported maltreatment in their own childhood.

### Methods

A protocol, based on the ENTREQ statement, was registered with PROSPERO. We searched Medline, PsycINFO, CINAHL, EMBASE, NHS Evidence and key Web of Science databases from date of inception to June 2018 to identify qualitative studies exploring perinatal experiences of parents who were maltreated in their own childhood. Two reviewers independently screened articles for inclusion and extracted data. Data were synthesised using grounded theory and thematic analysis approaches.

### Findings

The search yielded 18329 articles, 568 full text articles were reviewed, and 50 studies (60 articles) met inclusion criteria for this review. Due to the large number of studies across the

**Data Availability Statement:** All relevant data are within the manuscript and its Supporting Information files.

**Funding:** This review and the Healing the Past by Nurturing the Future project is funded by the Lowitja Institute Aboriginal and Torres Strait Islander Health CRC and the National Health and Medical Research Council (1141593). Catherine Chamberlain receives an NHMRC Career Development Fellowship (1161065). Stephanie Brown holds an NHMRC Senior Research Fellowship (1103976). Research staff at the MCRI are supported by the Victorian Government's Operational Infrastructure Support Program. The funders had no role in study design, data collection and analysis, decision to publish, or preparation of the manuscript.

**Competing interests:** The authors have declared that no competing interests exist.

whole perinatal period (pregnancy to two years postpartum), this paper reports findings for experiences during pregnancy, birth and early postpartum (27 studies). Parents described positive experiences and strategies to help them achieve their hopes and dreams of providing safe, loving and nurturing care for their children. However, many parents experienced serious challenges. Seven core analytic themes encapsulated these diverse and dynamic experiences: *New beginnings*; *Changing roles and identities; Feeling connected; Compassionate care; Empowerment; Creating safety;* and *Reweaving a future*.

## Conclusions

Pregnancy birth and the early postpartum period is a unique life-course healing opportunity for parents with a history of maltreatment. Understanding parent's experiences and views of perinatal care and early parenting is critical for informing the development of acceptable and effective support strategies.

## Introduction

Child maltreatment is experienced by 25–50% of children worldwide [1], and is a major public health challenge [2]. Child maltreatment can have profound and ongoing impacts on physical, social and emotional wellbeing [3, 4]. Increasing evidence demonstrates that cumulative traumatic experiences, particularly those involving interpersonal violation within a child's care network, can affect brain development [3] and activate conflicting attachment and defence systems [5]. This can lead to maladaptive behavioural and relational responses that are an attempt to manage distress but may also result in harm [3, 5] [6, 7].

Most current trauma research has focussed on post-traumatic stress disorder (PTSD) associated with exposure to events such as war, accidents and disaster. However, over two decades, the clinical literature has suggested that the diagnostic profile of PTSD is inadequate to capture the breadth and severity of symptoms related to repeated and prolonged child maltreatment [8]. There is currently growing international consensus around a distinct cluster of response or distress symptoms [7], and the 11th edition of the International Classification of Diseases (ICD-11) has recently formalised the diagnosis of Complex Posttraumatic Stress Disorder (CPTSD), or 'complex trauma'. However, this distinction has not yet been recognised in the Diagnostic and Statistical Manual of Mental Disorders (DSM-5) [9].

Complex trauma can also occur within the context of social institutions [10], and occur or be exacerbated by the cumulative effect of traumatic experiences as an adult. Given the relatively recent proposal and evolving consensus around the diagnostic profile, there are currently no clear prevalence estimates of complex trauma as a specific condition. The existing literature also uses a variety of measures and terms related to the common antecedent of complex trauma, exposure to child maltreatment (e.g. 'adverse childhood experiences'). In this review we use the term and focus on parents that have experienced 'child maltreatment' to reflect the existing literature, recognising the current dynamic state of definitions and terms being proposed. We use a definition of child maltreatment consistent with the World Health Organization (WHO) to include "all types of physical and/or emotional abuse and neglect, and sexual abuse that results in actual or potential harm of children in the context of relationships of responsibility, trust and power" [1].

Child maltreatment is strongly associated with other forms of adversity [11] [12] [6, 13]. A range of socio-ecological factors increase the risk of exposure to violence [14] [15]. Social

adversities interact with the ongoing effects of child maltreatment, with compounding inter-generational effects on health equities [16]. As such, this is a crucial human rights and global health issue to address for improving health equities worldwide. This evidence review is being conducted as part of a larger project in Australia, which aims to co-design perinatal awareness, recognition, assessment and support strategies for Aboriginal and Torres Strait Islander (Aboriginal) parents experiencing complex trauma [16]. Within the past two decades reports have documented high rates of child maltreatment and violence among Aboriginal communi-ties in Australia [17], and of infants being removed from Aboriginal families [18]. The reasons for this are complex and lie beyond the scope of this review but are a influenced by historical trauma including forced removal of children from family and culture, marginalisation and poverty [19, 20]. Prevention and early intervention strategies to address this crisis are urgently needed [21].

There are significant associations between child maltreatment and a wide range of physical and psychological health problems [22] [11, 23, 24] [23]. Critically, any long-lasting relational effects of childhood maltreatment may negatively impact on a parents ability to nurture and care for their children [5]. Emotional fear responses can be triggered by a range of experiences in pregnancy, birth and breastfeeding or the demands of early parenting, and may be experi-enced as conflicting feelings, rather than as thought-out narratives [25] [26]. Unresolved trauma is a key factor characterising difficult parent-child relationships and can severely dis-rupt development of capacities to manage internal states and self-regulate [3, 5].

Conversely, the transition to parenthood during the perinatal period (pregnancy to two years postpartum) offers a unique life-course opportunity for healing [27], [24] [28]. A positive focus during this has the potential to transform the 'vicious cycle' of intergenerational trauma into a 'virtuous cycle' that contains positively reinforcing elements that promote healing [28]. Fava et al. [28] found that post-traumatic changes during the perinatal period were more likely to be exclusively positive, whereas post-traumatic changes at other periods of the life course were likely to have more mixed positive and negative changes. A longitudinal study following youth after detention (who had experienced high rates of child maltreatment) [13] found that parenthood was one of the few 'turning points' that altered negative life course trajectories for women [29]. Despite these challenges, it is important to note that most parents who have expe-rienced childhood maltreatment demonstrate resilience and are able to provide safe and loving care for their children [30]. Understanding factors that shape these positive pathways are key to identifying innovative strategies for supporting parents with histories of child maltreatment [5].

'Life course approaches' are important for understanding and addressing health inequities [31]. The perinatal period is a particularly critical 'life course opportunity' where young people may be supported to process experiences of childhood maltreatment and/or complex trauma. There are increased risks of experiencing 'triggering' associated with intimate experiences of perinatal care and attachment demands of the new baby and other social health issues/stresses. Importantly, this is a critical opportunity for relational healing for the parent, and prevention of intergenerational transmission of trauma to the child. This is also the first time most people will have frequent and regular health care appointments as an adult.

Despite these specific risks and opportunities in the perinatal period, there is currently lim-ited specific guidance for trauma-informed perinatal care, few early parenting programs and no routine approaches to provide specific support for parents with a history of child maltreat-ment [32]. Australian Complex Trauma guidelines [10] provide direction on trauma-specific therapies, but there is limited guidance specifically in relation to parenting. Arguably, the need for trauma-informed care is particularly important during the perinatal period, where young parents may be engaging frequently with health systems for the first time since childhood; and

where the potential for triggering due to the intrusive nature of some perinatal procedures is highly salient.

Understanding perinatal experiences and views of parents who were maltreated in childhood, and may be experiencing complex trauma, is critical for contextualising and informing the development of safe and effective perinatal support strategies. This review has been developed to inform the co-design of perinatal awareness, recognition, assessment and support strategies for Aboriginal parents in Australia [16]. However, this systematic review is not limited to Aboriginal parents, as we know there would be no or very limited existing studies meeting this inclusion criteria. Additionally, while we recognise that the experience of child maltreatment is influenced by contextual and cultural factors, there are likely to be many relevant and shared individual experiences for this global health problem.

### Aim and review questions

This review aims to examine the pregnancy, birth and early postpartum experiences and views of parents who were maltreated in their own childhoods. The review questions are, during pregnancy, birth and the early postpartum period (up to six weeks after birth), for parents who report child maltreatment in their own childhoods:

1. What are their experiences of perinatal care?

2. What do they describe as barriers and enablers to improving access and quality of perinatal care?

3. How do they experience the transition to parenting?

4. What are their aspirations and challenges?

5. What strategies do they use and/or suggest might help or hinder healing and 'breaking the cycle' of maltreatment?

### Methods

This protocol has been developed with reference to the ENTREQ statement [33] and PRISMA-Equity guidelines [34] and registered on the PROSPERO database (CRD42018102110). See S1 Appendix for the PRISMA checklist completed for this review. The aims in the review protocol are to examine the *perinatal (pregnancy to two years after birth)* experiences and views of parents who were maltreated in their own childhoods. However, we retrieved a very large volume of eligible studies that were not amenable to meaningful qualitative synthesis (50 studies and theses reports). Therefore, we will report the perinatal experiences in two papers. This review includes studies exclusively or predominantly including parents during pregnancy, birth and early postpartum period (up to approximately six weeks after birth); or where the focus of the study was specifically on pregnancy, birth or breastfeeding. Studies focussing specifically on parenting, or involving parents exclusively or predominantly from the early postpartum period up to two years after birth, will be reported in a subsequent paper.

### Eligibility criteria

**Sample.** This review focuses on prospective and new parents (up to approximately six weeks after birth). However, the initial search included prospective and new parents (up to two years after birth), which was later refined as discussed below. As with a previous scoping review which informed the methods for this review [6], where mean ages only were reported (and it was unclear if children were two years of age or younger), studies reporting a mean age

of less than five years only were included. Studies in which any proportion of participants were parents of children aged two years or less were also included, to err towards inclusivity. However, we note that a number of studies included parents of children with varying ages. Due to the large volume of articles meeting these inclusion criteria, studies were then further categorised into those predominantly involving or focussing on parents views and experiences:

1. During pregnancy, birth and the early postpartum period (up to approximately six weeks after birth), for this paper.

2. From the early postpartum period up to approximately two years after birth, to be reported in a subsequent paper.

We used a broad definition of 'parent' and these could include same sex, adoptive or foster parents. We also included studies involving perinatal service providers caring for parents.

**Phenomena of interest.**   Experiences of perinatal care and the transition to becoming a parent (during pregnancy birth and the early postpartum period) for people who have experienced maltreatment in their own childhoods.

**Design.**   Studies were considered for inclusion if they utilised a qualitative study design and were published (or accepted) in a peer-reviewed journal. This included theses published in ProQuest Dissertation Abstracts International. Single case studies involving one or two parents were excluded, as these tended to have more of a focus on clinician observations, and the aim of this review is to understand parent views and experiences, and there was already sufficient data for saturation of thematic categories.

**Evaluation.**

- Parents' experiences of perinatal care and the transition to parenting.

- Any parent reports of challenges, aspirations or other issues.

- Reflections and perspectives of service providers about issues affecting parents.

**Research type.**   Qualitative research. There were no restrictions on settings or date, however only studies reported in English were included, owing to the language capacities of the review team.

## Identification of studies

**Sources.**   We searched for potentially relevant studies from databases and other sources from the date of database inception up to 22 June 2018. The following databases were searched: Medline (OVID), PsycINFO (OVID), CINAHL (EBSCO), EMBASE (OVID), several Web of Science databases (Social Sciences Citation Index, Book Citation Index (Social Sciences and Humanities), Emerging Sources Citation Index, Conference Proceedings Citation Index (Social Sciences and Humanities)) and NHS Evidence. These databases were selected to identify research from the fields of healthcare, health behaviour, nursing and social science, and include those that index both journal articles and research published outside journals. Articles identified from an earlier scoping review [6] (conducted partly to inform the design of the current review) were also screened for inclusion. Reference lists of similar reviews and included studies were checked for potentially relevant studies.

**Search strategy.**   We used a comprehensive search strategy that aimed to identify most of the relevant research in this area. Literature in this field uses diverse terminology and is not consistently indexed within bibliographic databases. The search was designed by an information specialist (CS) and the lead reviewer (CC) and aimed to balance sensitivity for identifying studies with the availability of information sources and resources for systematic screening. The

database searches were constructed around two concepts: (1) prenatal to postnatal care, transition to parenting, parenting or parents; and (2) a history of childhood maltreatment, post-traumatic stress or intergenerational trauma. Each concept comprised a range of terms and phrases to search the controlled indexing and title and abstract fields, in order to capture potentially-relevant studies that use a variety of terminology to describe the two concepts. The search was tested on Medline and PsycINFO and translated into other databases. The S2 Appendix presents a sample of the search strategy used with the PyscINFO database.

The search was informed by the over 50 research studies identified from the earlier scoping review [6], and additional studies located from 'related items' searching in PubMed and from three systematic reviews [35–37]. These studies were relevant to the topic area, but were not necessarily qualitative research. The index terms and words and phrases in the titles and abstracts were checked for relevant terms using text analysis and text mining tools. These were used to improve the sensitivity of the search and to prompt consideration of other alternative terms.

**Study screening and selection.** Electronic search results were imported into Endnote, de-duplicated and a copy of the original search saved. On a copied Endnote file, two reviewers independently screened all titles and abstracts against the selection criteria to identify potentially relevant studies. The full text of all potentially relevant studies was downloaded and further screened by two reviewers against clear selection criteria for inclusion. Discrepancies were resolved by consensus, and where necessary, a third reviewer. Articles were assessed to determine if they were publications from the same study, and categorised as 'primary', 'secondary' (if they belonged to the same study but included additional findings for line-by-line coding), or 'associated' if they didn't include relevant additional findings but may have other relevant information (e.g. protocols and methods papers).

**Data extraction.** Two reviewers independently extracted data related to the sample characteristics (including PROGRESS-Plus criteria, used to identify equity-relevant characteristics reported in studies) [38], study aims and scope, and risk of bias assessments into an Excel spreadsheet. These data were imported into NVivo 12 software [39] as file attributes, as presented in the S3 Appendix. All primary and secondary articles were imported into NVivo as files and linked to the relevant file attribute data. With the exception of descriptions of child maltreatment described by parents, which was not the focus of this study, all text under the 'Findings/Results' sections of all primary and secondary articles were coded line-by-line, as detailed below.

**Data synthesis.** An inductive grounded theory-based and thematic synthesis approach was used to synthesise study findings and assess concepts across included studies [40]. This involved thematic analysis techniques, including three stages of: line-by-line coding of text as axial codes; development of 'descriptive themes'; and generation of 'analytic themes'. We followed Noblit and Hare's seven-step process of: getting started; deciding what is relevant to the initial interest (research questions); reading the studies; determining how the studies are related; translating the studies into one another; synthesising translations; and expressing the synthesis [41]. Preliminary line-by-line coding was conducted in NVivo; three reviewers (CC/NR/SH) coded the first two studies together, and then a single reviewer coded a proportion of the remaining studies. Preliminary axial coding was discussed among the review author group to identify emergent themes and a process for developing descriptive themes.

Two reviewers worked together in 'shifts' (CC/NR/SH) to build familiarity with a smaller sample of studies and develop descriptive themes. This inductive process involved re-organising the study data and axial codes, and using a constant comparison method to discuss and record notes regarding emergent themes and explore both associations and exceptions to these themes. During synthesis, exceptions were explored to see if they could be explained by

attributes of the study population, such as type of child maltreatment (e.g. sexual vs physical abuse), gender of parent, young parental age (25 years or less), ethnicity, socio-economic or relationship status of the parent). Descriptive themes were then discussed with the broader review authorship team. Analytic themes were generated using visual mapping tools (see S4 Appendix for concept map) and were refined by the whole authorship team. Supporting parent and study author quotes for each axial code and/or descriptive theme are reported in the main text of the review findings. A summary table of analytic themes, descriptive subthemes and axial codes, including coding as *positive experiences*, challenges and *things that help* are outlined in the S5 Appendix.

**Critical appraisal of methodological quality within studies and overall confidence in findings from the synthesis.** Critical appraisal of methodological quality of individual studies were assessed using the Critical Appraisal Skills Programme (CASP) checklist for qualitative studies [42]. We made an overall judgement of each study (high, moderate, low or very low), based on whether there were serious concerns about the appropriateness of qualitative method, consideration of research-participant relationship, the sampling strategy, the method of data collection for addressing the research issue, the analysis approach and data richness.

Confidence in each of the themes arising from the meta-synthesis *across* studies was assessed using the GRADE-CERQual approach [43], based on consideration of four components: (1) methodological limitations of the studies contributing to each theme; (2) coherence of each theme (clear and cogent themes across studies); (3) adequacy of data supporting each theme (richness and quantity of data); and (4) relevance of the data from the studies supporting each theme to the review question. The review team considered these four components for the subset of studies contributing to each of the main 'analytic' themes emerging from the data [43] in order to generate a Summary of Qualitative Findings table [44]. Three reviewers (CC/NR/SH) also assessed confidence in each analytic theme for specific study attributes identified as important during synthesis, including child maltreatment other than sexual abuse; parent gender; young parental age; ethnicity; and relationship status of the parent. Confidence in findings when removing studies with most significant concerns regarding methodological quality (reflexivity) was also assessed. A table is presented in S6 Appendix that lists the analytic themes and descriptive subthemes by selected study characteristics. Assessment by study country was not explored as all but one study were conducted in high-income countries. Socio-economic status was also not assessed across each analytic theme as this characteristic was inadequately reported in included studies.

## Findings

### Results of search

The search yielded 18329 articles after de-duplication from all sources. 17590 articles were excluded on initial title/abstract screen, and a further 191 excluded following a second round of title/abstract screening conducted by two reviewers (CC/NR). The full text of 568 articles were assessed for eligibility and 503 were excluded (see S7 Appendix for table of excluded studies with reasons for exclusion). Fifty primary studies met the original inclusion criteria for the review related to parents across the entire perinatal period (65 articles, including 10 secondary articles and five associated articles). Of these, 27 primary studies (plus four secondary articles) predominantly included and/or reported parents' experiences and views during pregnancy, birth and the early postpartum period (up to approximately six weeks postpartum), and were included for the current review. See Fig 1 for search flow chart.

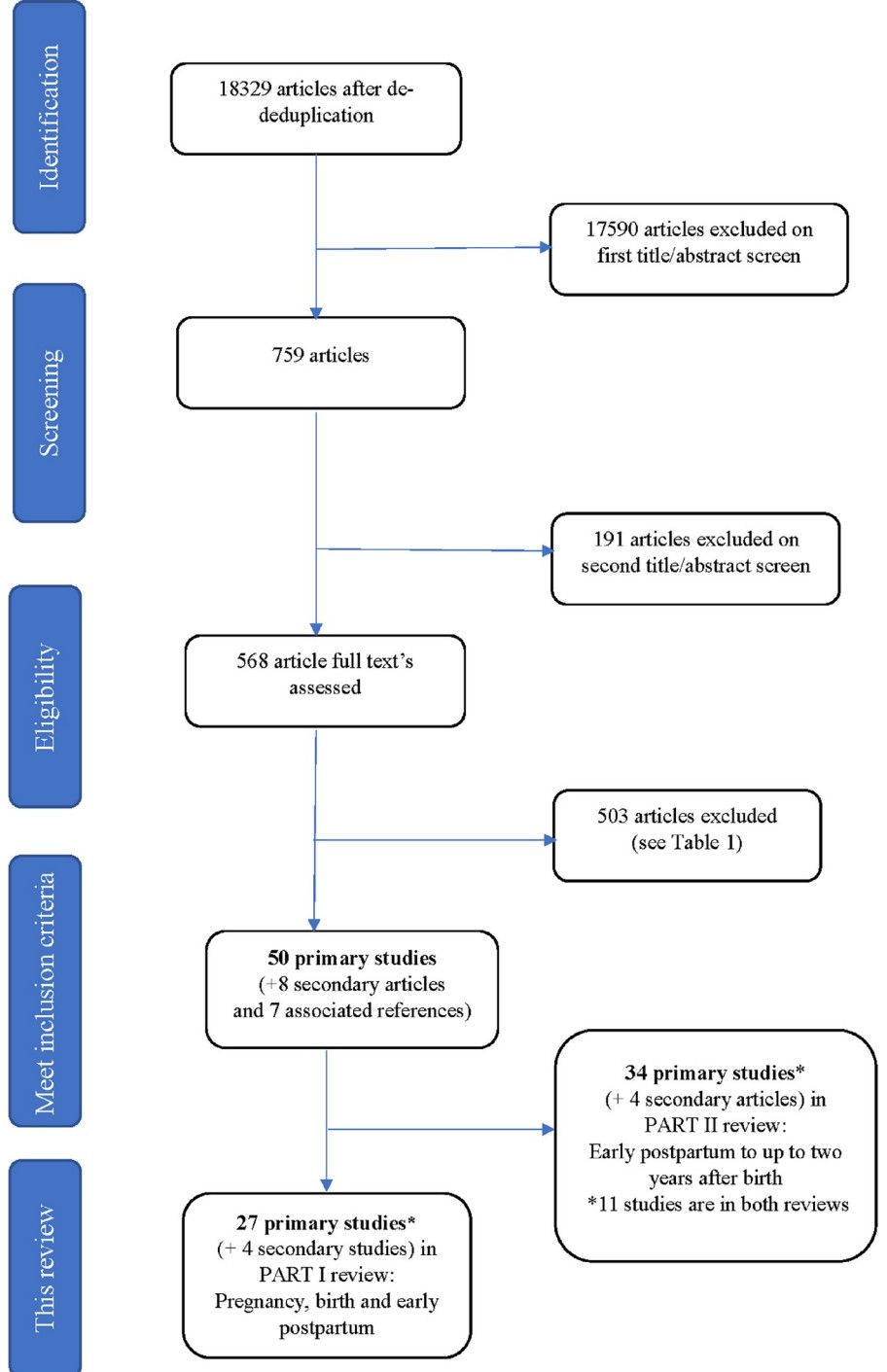

**Fig 1. Flow chart of included studies.**

## Characteristics of included studies

The characteristics of included studies are provided in Table 1.

**Setting.** The 27 studies included in this review were conducted in the United States of America (USA) (n = 16) [47, 52, 54, 55, 59, 60, 63–69, 71–73], United Kingdom (UK) (n = 4)

**Table 1. Characteristics of included studies.**

| Study ID | Country setting | Participants and setting | Perinatal period | Child maltreatment history | Study description | Data coded to analytic themes | Methodological quality summary |
|---|---|---|---|---|---|---|---|
| Berman et al., 2014 [45]* | Canada, Ontario | **32** women (34% ethnic and cultural minority, including 9% Indigenous) community-based urban | Pregnancy (first-time mothers; second trimester) | All reported child sexual abuse (CSA); interpersonal trauma (inclusion criteria) | Semi-structured interviews using thematic analysis (grounded theory) with a purposive sample exploring ways past interpersonal trauma shapes the transition to motherhood | 1. New beginnings<br>2. Changing roles and identities<br>3. Feeling connected<br>6. Creating safety<br>7. 'Reweaving' a future | High |
| Byrne et al., 2017 [46]* | UK | **3** women accessing mental health services | Pregnancy and parenting children 14 months to 23 years | All reported CSA (inclusion criteria); some child protection involvement | Open-ended interviews using a narrative research methodology examining the meaning women make of the impact of CSA on experiences of pregnancy, childbirth, and the postnatal period | 2. Changing roles and identities<br>3. Feeling connected<br>4. Compassionate care<br>5. Empowerment<br>6. Creating safety<br>7. Reweaving' a future | High |
| Cohen, 1987 [47]* | USA, north eastern city | **17** women (18% ethnic and cultural minority) accessing sexual assault centre services | Pregnancy and parenting children 9 months to 29 years | All reported CSA (incest; inclusion criteria); emotional and physical abuse and neglect and witnessing intimate partner violence (IPV) inferred from results | Sequential interview protocol with a self-selecting sample using thematic analysis (grounded theory) to examine mothering experiences of women who were incest victims, and the long range effects of this type of abuse | 2. Changing roles and identities<br>3. Feeling connected<br>5. Empowerment<br>6. Creating safety<br>7. 'Reweaving' a future | High |
| Coles & Jones, 2009 [48] (Coles, 2009 [49]) | Australia, Melbourne and Gippsland | **18** women community-based urban and rural | Interviewed at 6 weeks to 18 months postpartum, but data includes reflections on early breastfeeding experience. | All reported CSA (incest; inclusion criteria) | In-depth, semi-structured interviews with thematic analysis (grounded theory) with women with a history of CSA to explore responses to perinatal professional touch and examination of themselves and their babies; and their experience of successful breastfeeding | 2. Changing roles and identities<br>3. Feeling connected<br>4. Compassionate care<br>5. Empowerment<br>6. Creating safety<br>7. 'Reweaving' a future | High |
| Datta, 2017 [50]* | UK, England | **3** adolescent women taking part in support program for 'at risk' parents | Up to 12 months after birth, reflecting on birth experience | All had a 'looked after' care history (inclusion criteria) | Semi-structured interview using a Frameworks approach explores views and experience of women with care backgrounds of the group Family Nurse Partnership program | 4. Compassionate care<br>7. 'Reweaving' a future | Moderate |
| Garratt, 2018 [51] | UK, Britain | **8** women affiliated with CSA support groups and; **12** midwives who were also CSA survivor mothers | Interviewed from 19 months to 36 years after birth, about birth experience. | All reported CSA (inclusion criteria); emotional and physical abuse inferred from results | In-depth interviews using grounded theory techniques coupled with voice-centred relational approach to understand the problems and difficulties encountered by childbearing women who have a history of childhood sexual abuse | 2. Changing roles and identities<br>3. Feeling connected<br>4. Compassionate care<br>5. Empowerment<br>6. Creating safety<br>7. 'Reweaving' a future<br>Provider views | Moderate |
| Kennedy, 2010 [52]* | USA, large city | **14** adolescent women (100% ethnic and cultural minority; including 7% Indigenous) accessing homeless shelter and support program | Pregnancy (first time parents) and parenting (age not specified) | Reported CSA; physical abuse; community violence, witnessing and experiencing IPV; child protective services involvement and out-of-home placement. | Semi-structured interviews with a purposive sample using a grounded theory approach to understand violent victimization and adversity with mothers and how they adapt, shape their identity, and the role of social support | 1. New beginnings<br>3. Feeling connected | High |
| Lasiuk, 2007 [53]* | Canada, Edmonton | **7** women (43% Indigenous) community-based urban | Pregnancy and parenting children 12 months to 40 years | All reported CSA (inclusion criteria) | Interpretive inquiry using conversational interviews to explore the lived experience of pregnancy and birthing of women with histories of childhood sexual abuse | 1. New beginnings<br>2. Changing roles and identities<br>3. Feeling connected<br>4. Compassionate care<br>5. Empowerment<br>6. Creating safety<br>7. 'Reweaving' a future | High |

*(Continued)*

**Table 1.** (Continued)

| Study ID | Country setting | Participants and setting | Perinatal period | Child maltreatment history | Study description | Data coded to analytic themes | Methodological quality summary |
|---|---|---|---|---|---|---|---|
| Lee, 2001 [54] | USA | **7** women accessing perinatal care and mental health services | Parenting children 2 months to 6 years | All reported CSA (inclusion criteria) | Open ended interviews and thematic analysis (grounded theory) to examine ways being a survivor of sexual abuse interacts with the childbearing experience | 1. New beginnings<br>2. Changing roles and identities<br>3. Feeling connected<br>4. Compassionate care<br>5. Empowerment<br>6. Creating safety<br>7. 'Reweaving' a future | Moderate |
| McCoy, 2015 [55]* | USA, San Francisco | **6** adolescent women (100% ethnic and cultural minority) foster care leavers accessing care leavers support program | Parenting children 8 months to 6 years | All reported CSA (inclusion criteria); inferred child protective services involvement; other trauma not specified. | Using a phenomenological approach, questionnaire and semi-structured interviews focussed on pregnancy experiences and maternal perceptions of adolescent African American and biracial mothers with histories of sexual abuse | 1. New beginnings<br>2. Changing roles and identities<br>3. Feeling connected<br>4. Compassionate care<br>6. Creating safety<br>7. 'Reweaving' a future | High |
| Miura et al., 2018 [56]* | Brasil | **6** Brazilian young adolescent women in residential care | Pregnant and parenting (ages of children not given) | Intra-family violence (inclusion criteria); reported CSA, emotional and physical abuse and neglect, witnessing IPV and inferred child protective services involvement | Using semi-structured interviews and thematic analysis (grounded theory) to understand emotional experience from intra-family violence of institutionally sheltered pregnant and mothering adolescents | 1. New beginnings<br>3. Feeling connected<br>4. Compassionate care<br>6. Creating safety<br>7. 'Reweaving' a future | Moderate |
| Montgomery et al., 2015a [57] (Montgomery et al., 2015b [58]) | UK, south of England | **9** women accessing community-based maternity care | Interviewed 9 weeks to 28 years after birth, specifically regarding their maternity care experience | All reported CSA (inclusion criteria) | Narrative study using in-depth interviews subject to Voice-Centred Relational Method of analysis (and further thematic analysis / grounded theory) to explore the impact CSA has on the maternity care experiences of adult women to inform practice | 2. Changing roles and identities<br>3. Feeling connected<br>4. Compassionate care<br>5. Empowerment<br>6. Creating safety<br>7. 'Reweaving' a future | High |
| Muzik et al., 2013 [59] | USA | **52** women (39% ethic and cultural minority, including up to 10% Indigenous) accessing community-based obstetric care | Immediately after birth to 7 months postpartum | Reported CSA, emotional and physical abuse and neglect | Part of a mixed methods study, semi-structured interviews underwent content thematic analysis (grounded theory) to understand more about healthcare preferences of trauma-exposed women in the early postpartum period | 1. New beginnings<br>2. Changing roles and identities<br>4. Compassionate care<br>5. Empowerment<br>6. Creating safety<br>7. 'Reweaving' a future | Moderate |
| O'Brien, 1999 [60]* | USA, Norfolk, Virginia | **9** women (44% ethnic and cultural minority) accessing mental health services | Interviewed 5 months to 18 years after birth, including reflections on pregnancy and birth | All reported CSA (incest; inclusion criteria), emotional and physical abuse and neglect inferred from results | In-depth interviews with purposive community sample using constant comparative method (grounded theory) to focus on parenting by incest survivors, and the complex interactions between mother and child | 1. New beginnings<br>2. Changing roles and identities<br>3. Feeling connected<br>5. Empowerment<br>7. 'Reweaving' a future | High |
| Palmer, 2005 [61]* | Canada, Vancouver | **46** women (28% ethnic and cultural minority, including 24% Indigenous) accessing CSA support services and community-based urban; **22** health care professionals | Pre-conception, pregnancy and post-partum up to 12 months, to parenting up to 10 + years, and childless by choice | All reported CSA (inclusion criteria); emotional and physical abuse, witnessing community violence and IPV inferred from results | Interviews and focus groups using constant comparative method (grounded theory) to understand the experiences of childbearing women who are childhood sexual abuse survivors and their health care providers, to generate a theoretical model explicating the ways they manage, negotiate, or realise their childbirth experience | 1. New beginnings<br>2. Changing roles and identities<br>3. Feeling connected<br>4. Compassionate care<br>5. Empowerment<br>6. Creating safety<br>7. 'Reweaving' a future | High |

(*Continued*)

**Table 1.** (Continued)

| Study ID | Country setting | Participants and setting | Perinatal period | Child maltreatment history | Study description | Data coded to analytic themes | Methodological quality summary |
|---|---|---|---|---|---|---|---|
| Parratt, 1994 [62] | Australia, central Victoria | **6** women accessing counselling and incest support groups | Parents interviewed 6 months to 21 years after birth, specifically regarding childbirth experience. | All reported CSA (incest; inclusion criteria) | In-depth interviews using a phenomenological approach to discover the experiences, including feelings, women who are survivors of incest have during childbirth | 2. Changing roles and identities 3. Feeling connected 4. Compassionate care 5. Empowerment 6. Creating safety 7. 'Reweaving' a future | Moderate |
| Rhodes & Hutchinson, 1994 [63] | USA, Florida | **7** women accessing community-based perinatal care; **8** nurses | Childbirth | All reported CSA (inclusion criteria) | Ethnographic method and in-depth interviews used to explore labour experiences and perceptions of CSA survivors, and their caregivers (nurse-midwives and nurses) | 5. Empowerment 7. 'Reweaving' a future | Low |
| Richmond, 2006 [64] | USA, San Diego, California | **11** women (64% ethnic and cultural minority) accessing community-based perinatal care and counselling services | Childbirth. Within 1 year of birth | All reported CSA (inclusion criteria) | In-depth interviews and constant comparative analysis (grounded theory) to explore the meaning having a history of CSA has to pregnant survivors, and their perceptions of perinatal care | 1. New beginnings 2. Changing roles and identities 3. Feeling connected 4. Compassionate care 5. Empowerment 6. Creating safety 7. 'Reweaving' a future | High |
| Roberts, 2011 [65]* | USA, Pennsylvania | **8** parents (6 women, 2 men; 13% ethnic and cultural minority) accessing mental health services | Parenting (ages of children not given) | Childhood maltreatment (inclusion criteria) included sexual, emotional and physical abuse, and neglect not otherwise specified | Interviews and notes / journal underwent a constant comparative analysis (grounded theory) to explore the process of becoming a parent for adults who experienced maltreatment in childhood | 1. New beginnings 2. Changing roles and identities 3. Feeling connected 5. Empowerment 6. Creating safety 7. 'Reweaving' a future | High |
| Roller, 2011 [66] | USA, large Midwest city | **12** women (75% ethnic and cultural minority) accessing community-based perinatal care | Pregnancy; post-partum to 12 months and parenting (ages of children not given) | All reported CSA (inclusion criteria) | Open-ended interviews using a constant comparative analysis (grounded theory) to construct a theoretical framework to describe how CSA survivors manage the pain of intrusive re-experiencing of CSA during the perinatal period | 4. Compassionate care 5. Empowerment 6. Creating safety 7. 'Reweaving' a future | Moderate |
| Saewyc, 2000 [67] | USA, Seattle | **8** adolescent women (38% ethnic and cultural minority, including 13% Indigenous) living out-of-home (inclusion criteria) attending youth perinatal care and housing services | Pregnancy; post-partum to 4 weeks | Reported CSA, emotional and physical abuse and neglect; child protective services involvement; witnessing IPV; sexual violence and IPV. | Methods adapted from feminist anthropology guided ethnographic interviewing and participant observation to explore meanings attributed to pregnancy and parenting by out-of-home pregnant adolescent women during the course of pregnancy and early postpartum | 1. New beginnings 2. Changing roles and identities 3. Feeling connected 6. Creating safety 7. 'Reweaving' a future Provider views | High |
| Schwerdtfeger & Wampler, 2009 [68] | USA, Oklahoma | **10** women (10% ethnic and cultural minority) accessing community-based perinatal care | Pregnancy | Interpersonal sexual trauma (inclusion criteria), CSA abuse and physical abuse | Semi-structured interviews and a questionnaire with phenomenological perspective to explore the lived experience of pregnant women with histories of sexual trauma | 1. New beginnings 3. Feeling connected 5. Empowerment 6. Creating safety 7. 'Reweaving' a future | High |
| Seng et al., 2002 [69] (Seng et al., 2004 [70]) | USA, Iowa, Michigan | **15** women (13% ethnic and cultural minority) community-based urban and rural | Post-partum 1 week to parenting children up to 26 years | All reported CSA (inclusion criteria) | Using a narrative analysis approach, interviews with women with a history of CSA and abuse-related posttraumatic stress during the childbearing year to determine perceptions of optimal maternity care | 3. Feeling connected 4. Compassionate care 5. Empowerment 6. Creating safety 7. 'Reweaving' a future | Moderate |

(*Continued*)

**Table 1.** (Continued)

| Study ID | Country setting | Participants and setting | Perinatal period | Child maltreatment history | Study description | Data coded to analytic themes | Methodological quality summary |
|---|---|---|---|---|---|---|---|
| Swartz et al., 2012 [71] | USA, southwest city | **10** women and **8** male partners (44% ethnic and cultural minority, including 10% Indigenous) accessing local community agencies | Pregnancy | Reported CSA, emotional and physical abuse (inclusion criteria); witnessing IPV | Semi-structured clinical interview underwent thematic analysis (grounded theory) to examine perceived influence of history of childhood abuse on parenting among pregnant couples | 1. New beginnings 2. Changing roles and identities 3. Feeling connected 6. Creating safety | Very low |
| White et al., 2016 [72] | USA | **6** women (100% ethnic and cultural minority) accessing trauma support services | not specified | Reported child sexual, emotional and physical abuse (inclusion criteria), neglect, and neglect not otherwise specified | Focus group using thematic analysis (grounded theory) to explore abuse survivor perspectives on optimal physician approaches to trauma inquiry in prenatal care | 4. Compassionate care 5. Empowerment 7. 'Reweaving' a future | Moderate |
| Williams & Vines 1999 [73] | USA | **7** adolescent women attending support program for 'at risk' parents | Post-partum 2 weeks to 3 months | Reported CSA, emotional and physical abuse (inclusion criteria) and neglect not otherwise specified, and child protective services involvement | Uses a phenomenological approach with an open-ended interview guide to uncover mother's views of their childhood; transition to motherhood; their important relationships after they became parents; and the positive impact parenting has | 1. New beginnings 3. Feeling connected 4. Compassionate care 7. 'Reweaving' a future | Moderate |
| Wood & van Esterik, 2010 [74] | Canada, Saskatoon, Sask | **6** women staff and volunteers at a healing centre for CSA | Not specified | All reported CSA (inclusion criteria) | A qualitative, participatory study using semi-structured in-depth interviews and thematic analysis (grounded theory) to explore infant feeding experiences and decisions of mothers who had been sexually abused as children | 2. Changing roles and identities 3. Feeling connected 4. Compassionate care 5. Empowerment 7. 'Reweaving' a future | High |

*Also meets inclusion criteria for paper including parents from early postpartum up to two years after birth.

CSA = child sexual abuse; IPV = intimate partner violence.

[46, 50, 51, 57], Canada (n = 4) [45, 53, 61, 74], Australia (n = 2) [48, 62] and Brazil (n = 1) [56].

Participants were recruited from a range of settings, and in 17 studies these were classified as 'risk' settings, including mental health settings (n = 4) [46, 54, 60, 65], trauma, sexual assault or incest support groups (n = 6) [47, 51, 61, 62, 72, 74], adolescent out of home care support or residential care programs (n = 3) [55, 56, 67], support programs for 'at risk' parents (n = 2) [50, 73], local community agencies such as perinatal care Special Supplemental Nutrition Program for Women, Infants, and Children, Craigslist and hospitals [71] and a homeless shelter program [52]. Other settings were classified generally as community-based urban [45, 53], urban and rural [48, 69], and perinatal, maternity or obstetric care [57, 59, 63, 64, 66, 68].

**Participants.** Three hundred and fifty-one parents participated in the included qualitative studies. Most (n = 25) studies included mothers exclusively, with only two studies involving fathers, (n = 2 and n = 8 fathers, respectively) [65, 71]. In addition to parents' views, three studies also incorporated the views of 12 midwives who were also survivors of childhood sexual abuse [51], 22 health care professionals [61] and eight nurses [63].

The ages of parents ranged from 13 to 60 years, with six studies including exclusively adolescent parents less than 25 years of age [50, 52, 55, 56, 67, 73]. Nineteen studies reported the race/ethnicity of parents, and in five of these, more than 50% of the participants identified as a member of an ethnic minority group [52, 55, 64, 66, 72]. Seven studies included a small

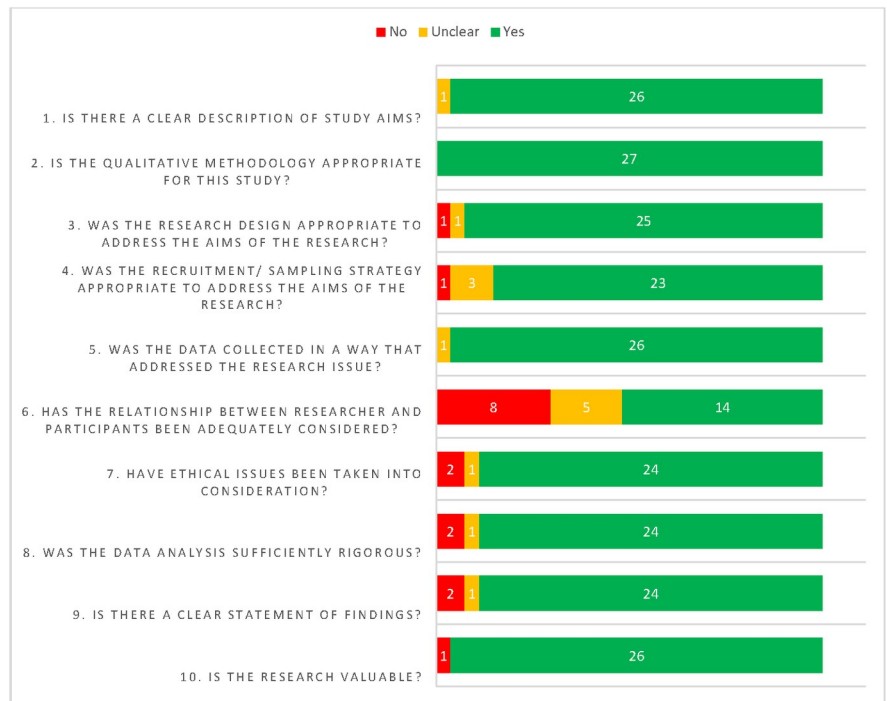

**Fig 2. Summary of critical appraisal assessments.**

proportion of Indigenous parents [45, 52, 53, 59, 61, 67, 71]. There were approximately 20 Indigenous parents in total, but we were unable to specify the exact number as Indigenous parents were combined with Asian parents in one study [59].

Of the 21 studies reporting parents' relationship status, eight identified the majority of parents as 'single' (>50%) [45, 50, 52, 56, 60, 61, 66, 73], and six identified a high proportion of single parents (>20% of the sample). Education and socio-economic status were reported in less than 15 studies and so are not reported here.

All studies in this review included some (n = 9) or all (n = 18) parents who reported experiencing childhood sexual abuse. In nine studies, a proportion of participating parents reported experiencing childhood sexual abuse [50, 52, 56, 59, 65, 67, 71–73]. In the remaining 18 studies, all parents participating in the study reported experiencing childhood sexual abuse. Other types of abuse that were described in this review included emotional and physical abuse; neglect; children experiencing intimate partner violence (IPV) or other violence occurring in the childhood home; and removal of children from family of origin. Though the type of abuse is not explicit, we considered removal from family of origin as a proxy measure for child maltreatment, as this is the most common reason for removal.

**Study design.** Qualitative studies included in this review collected data using open or semi-structured individual interviews in most (n = 24) studies. One study used ethnographic observation [63] and another also used questionnaires in addition to individual interviews [68]. One study collected data in a focus group setting [72]. Studies used grounded theoretical, narrative, phenomenological, framework, ethnographic or interpretive enquiry approaches to explore and analyse the data.

**Critical appraisal of the methodological quality of studies.** The majority of studies were judged to be of high methodological quality for most items on the CASP checklist [42] (Fig 2). Consideration of the relationship between the researcher and participants (item 6) was the

exception (not discussed in eight studies [50, 52, 59, 62, 63, 66, 71, 73] and unclear in five studies [48, 57, 60, 68, 74]). There is always a concern with research that participants may feel reluctant or unable to be frank or open about the challenges they face or they might provide socially desirable responses because of the sensitive nature of the topic. However, this was not rated as serious concern in many of these studies due to the richness and frankness of data presented. Of the 27 studies, 15 were rated as of high methodological quality overall (i.e. no or very minor concerns for any of the CASP items), 10 as moderate (due to moderate to serious concerns about consideration of research-participant relationship [50, 59, 62, 66], adequacy of data [54, 56], unclear sampling [69, 72, 73], or synthesis/analysis of data [51]), one as low and one as very low (Table 1). The study rated as low [63] had limitations in the sampling strategy, consideration of researcher-participant relationship and data adequacy. The study rated as very low [71] had limitations in research design and consideration of researcher-participant relationship. Specific assessment details for individual studies are available on request. A summary critical appraisal assessment of overall methodological quality for each study, as described in the methods, is provided in Table 1.

## Meta-synthesis of study findings

**Summary of findings.**   Parents in this review described *positive experiences* and hopes for the future associated with becoming a parent. However, they also described many *challenges* at multiple levels, including internal distress, interpersonal relationships, interactions in the community, and broader societal factors. Importantly, parents described many *things that help*, which parents are currently using and feel will help them to heal and nurture their family.

The following *seven broad analytic themes* emerged from the grounded theory and thematic analysis, relating to parents' experiences during pregnancy, birth and early postpartum:

1. *New beginnings*: Becoming a parent is an opportunity for 'a fresh start', to put the past behind them and move forward with hope for the future to create a new life for themselves and their child.

2. *Changing roles and identities*: Becoming a parent is a major life transition, influenced by perceptions of the parenting role.

3. *Feeling connected*: The quality of relationships with self, baby and others has major impacts on the experiences of becoming a parent.

4. *Compassionate care*: Kindness, empathy and sensitivity enables parents to build trust and feel valued and cared for.

5. *Empowerment*: Control, choice and 'having a voice' are critical to fostering safety.

6. *Creating safety*: Parents perceive the 'world as unsafe' and use conscious strategies to build safe places and relationships to protect themselves and their baby.

7. *'Reweaving' a future*: Managing distress and healing while becoming a parent is a personal ongoing and complex process requiring strength, hope and support.

Our confidence in each of the analytic themes is high based on the GRADE CERQual assessment (summarised in Table 2). Removing studies with low reflexivity (the main methodological limitation found in included studies) did not influence our confidence. However, we did have minor or moderate concerns regarding data adequacy and relevance when considering available studies for some subpopulations, which reduced our confidence in some findings for fathers, young parents (less than 25 years of age), parents from ethnic minority groups,

**Table 2.  Summary of findings and assessment of confidence in analytic themes using GRADE CERQual.**

| Analytic Theme | Studies contributing to finding | Methodological limitations | Coherence | Adequacy | Relevance | CERQual rating | Comments |
|---|---|---|---|---|---|---|---|
| *New beginnings*: Becoming a parent is an opportunity for '*a fresh start*', to put the past behind them and move forward with hope for the future to create a new life for themselves and their child | [45, 52–56, 59–61, 64, 65, 67, 68, 71, 73] | Very minor concerns about methodological limitations related to reflexivity. Removal of seven articles where consideration of researcher-participant relationship (reflexivity) was not reported did not change this finding. | No or very minor concerns about coherence across studies. | No or very minor concerns about data adequacy related to mothers with a history of Child sexual abuse (CSA) and/or other forms of child maltreatment (CM). Some or moderate concerns [few studies] for: fathers, and ethnic minority parents. | No or minor concerns about relevance related to mothers. Moderate concerns for: fathers, and ethnic minority parents. | **High Confidence** | These findings mainly come from a population of mothers from an ethnic majority with a history of CSA and/or other forms of CM. We are much less confident about these findings for fathers and ethnic minority parents. |
| *Changing roles and identities*: Becoming a parent is a major life transition, influenced by perceptions of the parenting role. | [45–49, 51, 53–55, 57–62, 64, 67, 71, 74] | Very minor concerns about methodological limitations related to reflexivity. Removal of nine articles where consideration of researcher-participant relationship was not reported did not change this finding. | Very minor concerns about coherence across studies. Reflecting the theme, parents' experiences and views related to roles and identities varied across time and between parents. | No or very minor concerns about data adequacy related to mothers with a history of CSA and/or other forms of CM. Some or moderate concerns [few studies] for: fathers, single parents, ethnic minority parents. | No or very minor concerns about relevance for mothers who have experienced CSA and/or other forms of CM. Moderate concerns for: fathers, single parents, and ethnic minority parents. | **High confidence** | These findings mainly come from a population of mothers from an ethnic majority with a history of CSA and/or other forms of CM. We are much less confident about these findings for fathers, single parents and ethnic minority parents. |
| *Feeling connected*: The quality of relationships with self, baby and others has a major impact on the experience of becoming a parent. | [45–49, 51–58, 60–62, 64, 67–71, 73, 74] | Very minor concerns about methodological limitations related to reflexivity. Removal of ten articles where consideration of researcher-participant relationship was not reported did not change this finding. | Very minor concerns about coherence across studies. Across and within studies, there are varied and diverse experiences between parents. | No or very minor concerns about data adequacy related to mothers with a history of CSA. Minor to moderate concerns for; parents who have experienced other types of CM, fathers, young parents, single parents, and ethnic minority parents. | No or very minor concerns about relevance for mothers who have experienced CSA. Minor to moderate concerns for: parents who have experienced other types of CM, fathers, young parents, single parents, and ethnic minority parents. | **High confidence** | These findings mainly come from a population of mothers from an ethnic majority with a history of CSA. We are less confident about these findings for parents with other types of CM, and much less confident about these findings for fathers, young parents, single parents and ethnic minority parents. |
| *Compassionate care*: Kindness, empathy and sensitivity enables parents to build trust, feel valued and cared for. | [46, 48–51, 53, 54, 56, 57, 59, 61, 62, 64, 66, 69, 72, 74] | Minor concerns about methodological limitations related to reflexivity. Removal of ten articles where consideration of researcher-participant relationship was not reported did not change this finding. | Very minor concerns about coherence across studies. | No or very minor concerns about data adequacy related to mothers with a history of CSA. Minor concerns for: parents who have experienced other types of CM and young parents. | No or very minor concerns about relevance for mothers who have experienced CSA. Minor concerns about data relevance related to parents who have experienced other types of CM, and young parents. | **High confidence** | This theme relates to perinatal care, and it is not expected that fathers would necessarily be represented here. These findings mainly come from a population of mothers with a history of CSA. We are much less confident about these findings for parents who have experienced other types of CM and young parents. |

*(Continued)*

**Table 2.** (Continued)

| Analytic Theme | Studies contributing to finding | Methodological limitations | Coherence | Adequacy | Relevance | CERQual rating | Comments |
|---|---|---|---|---|---|---|---|
| *Empowerment*: Control, choice and 'having a voice' are critical to fostering safety. | [46–48, 51, 53, 54, 56–58, 60–66, 68–70, 72, 74] | Very minor concerns about methodological limitations related to reflexivity. Removal of nine articles where consideration of researcher-participant relationship was not reported did not change this finding. | Very minor concerns about coherence across studies. Across and within studies, there are varied and diverse experiences between parents | No or very minor concerns about data adequacy related to mothers with a history of CSA. Moderate concerns for: fathers, parents who have experienced other types of CM and young parents. Minor concerns for: ethnic minority parents, single parents. | No or very minor concerns about relevance for mothers who have experienced CSA. Moderate concerns for: fathers and parents who have experienced other types of CMt, and young parents. Minor concerns for: ethnic minority parents, and single parents. | **High confidence** | These findings mainly come from a population of mothers with a history of CSA. We are much less confident about these findings for fathers and parents who have experienced other types of CM, and young parents. We are less confident about these findings for ethnic minority parents, and single parents. |
| *Creating safety*: Parents perceive the 'world as unsafe' and use conscious strategies to build safe places and relationships to protect themselves and their baby. | [45–48, 51, 53–57, 59, 61, 62, 64–68, 71] | Very minor concerns about methodological limitations related to reflexivity. Removal of seven articles where consideration of researcher-participant relationship was not reported did not change this finding. | Very minor concerns about coherence across studies. | No or very minor concerns about data adequacy related to mothers with a history of CSA and/ or other forms of CM. Moderate concerns for: young parents. | No or very minor concerns about relevance for mothers who have experienced CSA and/or other forms of CM. Moderate concerns for: young parents | **High confidence** | These findings mainly come from a population of mothers with a history of CSA and/ or other forms of CM. We are much less confident about these findings for young parents. |
| *Reweaving a future*: Managing distress and healing while becoming a parent is a personal ongoing complex process requiring strength, hope and support. | [45–48, 50, 51, 53–55, 57–70, 72–74] | Very minor concerns about methodological limitations related to reflexivity. Removal of eleven articles where consideration of researcher-participant relationship was not reported did not change this finding. | Very minor concerns about coherence across studies. Across and within studies, there are varied and diverse experiences between parents that reflect the different stages of recovery. | No or very minor concerns about data adequacy related to mothers with a history of CSA and/ or other forms of CM. Moderate concerns related to adequacy of data for fathers. | No or very minor concerns about relevance for mothers who have experienced CSA and/or other forms of CM. Moderate concerns about relevance related to fathers. | **High confidence** | These findings mainly come from a population of mothers with a history of CSA and/ or other forms of CM. We are much less confident about these findings for fathers. |

CSA = child sexual abuse; CM = child maltreatment; ethnic minority parents = categorised as the majority belonging to an ethnic minority group; young parents = all parents < 25 years.

single parents, and parents with a history of child maltreatment other than child sexual abuse (see Table 2 and S6 Appendix).

Analytic themes were often interrelated and mutually reinforcing and linked to other sub-themes (see concept map of generated themes in S4 Appendix). However, parents' experiences are personal and varied. A diverse range of experiences were reported 'between' parents, as well as internal conflicting experiences 'within' parents, which are dynamic and changing over time. The seven themes encompass the diversity of experiences, reasoning, emotions and behaviours reported by parents, both encouraging and challenging, as well as the strategies and resources that may support healing and improve perinatal experiences. Analytic themes, descriptive themes and axial codes are summarised in a table in the S5 Appendix, and are

described in detail with supporting quotes below. A more comprehensive, long version of the set of quotes for each theme are provided in S8 Appendix. Axial codes are highlighted in **bold**. Supporting quotes from parents (primary level data) are indented and provided in "*10 font italics*", while supporting quotes from article authors (secondary level data) are presented in "plain 12 font".

### Theme 1. *New beginnings*: Becoming a parent is an opportunity for 'a fresh start', to put the past behind them and move forward with hope for the future to create a new life for themselves and their child

This analytic theme incorporates three main descriptive subthemes; *new opportunities and motivation to change*; *hopes and dreams for the future*; and *wanting to parent differently*.

**1.1 New opportunities and motivations for change.** For many participants, pregnancy and the transition to parenthood was a turning point in their life trajectory, offering a **new opportunity or fresh start** for positive change or 'second chance' and a "*normal*" life that was distinct from their past trauma [45, 53–55, 59, 61, 65, 67, 68, 73]. This links to *the healing journey though pregnancy, birth and parenting* (subtheme 7.4).

> "*I was just happy, like oh my gosh! I didn't think of it as my world was ending, you know, or life was ending, I thought of it, you know, as a new beginning for me and my child.*" [67]

The confirmation of pregnancy elicited a new sense of responsibility in many expectant mothers to practice **self-care or take care of self for baby** and lead a healthier lifestyle [53, 54, 64].

> "*So I started taking better care of myself, because I was very motivated to make sure she started out okay and that she had me there, and I had to be as close to full capacity as I could get, so I could be there for her.*" [54]

In some studies, pregnancy provided women the motivation to step away from risky behaviours including drug and alcohol use, and the opportunity for **maturing or changing behaviour for the baby**, such as "*settling down*" by finding a steady job, returning to school, and accessing services and stable housing [53, 61, 67].

**1.2 Hopes and dreams for the future.** Compared to participants' past trauma experiences, pregnancy inspired new **hopes or dreams for the future** [45, 52, 59, 67, 68, 73]. These feelings of hope and optimism were also common among expectant mothers facing difficult circumstances (i.e. single, homeless or adolescent mothers) and were embedded in their motivation to create a **stable home life or family** [45, 52, 53, 65, 67] which differed from their own childhood. Parents also reported a **stable or safe relationship** as an important part of this [53]. This subtheme links to *wanting to parent differently* (subtheme 1.3).

> "*I wanted stability. I want a safe environment. Not just a safe home, but a safe community . . . a safe environment . . . a safe, happy, healthy environment and not just but for him and me, but for our families as well.*" [53]

Expectant adolescent mothers **hoped their partner would be a good father** [67], and shared **hopes for their child** [55, 68] regarding their appearance, personality and academic achievements.

> "*I imagined him like, I had this dream one night I had already had him and his older brother. They were all grown up and doing stuff, graduated from high school, went to college. Did what*

*they were supposed to do. Got great jobs. Had a family and then you know, I was peaceful and you know I died in my sleep. It was peaceful to actually see that.*" [55]

Women with a history of sexual trauma reported hopes that their child would never experience sexual abuse.

"*I hope and pray things don't happen to her the way that they did [for me] so that she will have a good life.*" [68]

In describing their hopes and desires, some parents expressed highly idealised notions of **a 'perfect' or idealised family** that was distinctly different from their own upbringing [45, 61]. Parents reflected on and 'self-evaluated' their parenting, with some suggesting an ideal family would have finely balanced and supportive relationships; a perfect understanding between mother and partner, and mother and child; a sense of stability and trust; and be free of violence [45, 61]. Pregnant women also highlighted the importance of **being a 'good mother'** [45, 53, 54, 67], which was again related to their own unmet needs during childhood, and links to *striving to be a 'good' or 'perfect' parent* (subtheme 2.2). Being a 'good mother' involved being present and focussed on the future, not the past, and 'moving on' [49].

"*And I think a lot of that was motivated by what happened to me. And I spent a lot of time thinking about what it meant to be a mother. I spent a lot of time reading about it as best I could, to try to understand it before I had a daughter. And because I really wanted to be a good mother.*" [54]

Focusing on the future and being a 'good mother' were key to making positive life changes.

"*If you want to change your life you, you have to think about your child, about your happy family. And what a good mother you're going to be. Concentrate on [the] positive. The past is past. I decided to live with my present, and just look into the future. And there is no bad in the future for me, I know.*" [45]

**1.3 Wanting to parent differently.** Enmeshed in parents' hopes and dreams for the future was an intense **desire to not repeat the past** [53, 56, 61, 67, 71] and **wanting to parent differently** [45, 53, 61, 64, 65, 71] so their children could have a life unlike their own.

"*From the point when I started trying to conceive I was thinking about the baby. I was thinking how good is going to be his or her life. Not like mine.*" [45]

Mothers and fathers described turning their past abuse into a "*positive*" [71]. Some felt their experiences taught them what not to do as they approached parenthood and instilled a desire to not repeat the past and break the cycle of abuse. However, parents described a lack of role models and discussed more about 'what *not* to do' rather than 'what to do'.

"*[My mum] put a lot things in perspective for me. I learned what not to do, who not to be, and how not to act and what not to do to my own kids, which I kind of am thankful for.*" [71]

"*I really want to raise my boys differently. Stop the cycle of abuse before it begins again.*" [61]

Parents also described challenges of feeling a **fear of repeating the past** [53, 55, 56, 59, 61, 64, 71], impacting on early attachment, which links to subthemes 6.1 and 7.1 regarding *safety*, *fear* and *trust*.

*"I had gotten pregnant with my second son, I was so afraid. I never sought therapy to sort myself out, but I was so afraid I was going to end up like my mother. And, or end up with somebody like my father or whatever. But, that just terrified me and the second time I got pregnant I thought about having an abortion cause I thought I was doing okay with one, but I didn't want to have that added stress, you know?"* [64]

Parents reflected on the type of abuse they had experienced (e.g. verbal abuse, physical abuse, exposure to IPV) and resolved to never inflict that upon their own child [53, 56, 67, 71]. However, one study found some expectant mothers and fathers with a history of childhood physical abuse did intend to physically discipline their child (e.g. spanking) [71].

In several studies, parents talked about **wanting to be emotionally responsive** and caring with their child, and connect with their children as they grow up, often because they felt their own parents had been emotionally unavailable [53, 55, 60, 67, 68].

*"There was really nowhere that I felt I was safe and where I was loved just for me. Now I have my own my family, this is my chance to make a safe space not just for my daughter, but for me and her father, and to create memories that are happy and healthy . . . not sort of tainted."* [53]

### Theme 2. *Changing roles and identities*: Becoming a parent is a major life transition, influenced by perceptions of the parenting role

This analytic theme included several descriptive themes around: *mixed emotions in pregnancy and birth*; *striving to be a 'good' or 'perfect' parent*; *wanting to be 'normal'*; and *knowledge and learning about parenting*.

**2.1 Mixed emotions in pregnancy and birth.** Mothers shared a range of **conflicting internal emotions about pregnancy** [45, 51, 53, 55, 58, 60, 61, 65, 67]. These experiences link to *relationships with self/body* (subtheme 3.2), and a **lack of control** (see subtheme 5.1). Many described feelings of excitement, elation and joy in **becoming pregnant** [47, 53, 55, 60, 67]. However reactions were also often mixed, with conflicting emotions of **hopes and fears** [55] for the future, as well as excitement and uncertainty, worry, nervousness or fear as expectant mothers considered the impending responsibility of parenthood [51, 53, 55, 60, 67]. These mixed feelings were often reported by young or single mothers, women with an unplanned pregnancy and women with fertility concerns. Feelings of shame or embarrassment were common among pregnant adolescents [53, 55, 60]. Some women also described the **impact of pregnancy on their current lifestyle** [53, 67] as a challenge. Some women reflected on their past trauma experiences when describing their ambivalent feelings about becoming pregnant [45, 53, 60, 61].

*"I was very anxious about what I could do to a baby because I was so mixed up and distraught on so many different levels. At some point, there was—it was almost like there was an answering consciousness—giving me a strong message that everything was going to be okay. I settled down and I started to think about whether it was a boy or a girl and started to engage with my pregnancy."* [53]

One parent noted that the happiness she felt contributed to her ambivalence about her pregnancy.

*"I was actually pretty happy. It was the first time that I ever felt good about myself. It's hard to feel good. Me feeling good means that I have to be punished. Feeling good is not comfortable. It is extremely uncomfortable."* [60]

For many mothers, feelings about becoming pregnant were strongly influenced by the quality of the *relationship with their partner* (subtheme 3.4), as to whether the experience of becoming pregnant was described in a positive or negative way. Other family members reactions to the pregnancy also had a significant impact on their experience, with reactions of adolescent parents' family members more mixed and associated with feelings of **stigma and judgement** (see subtheme 6.2). **Postpartum adjustment** [61] was a challenge for some parents who were dealing with triggers and often with very limited support systems.

"*It [postpartum period] was such a difficult time for me. I just remember it being a really black place . . . almost like I was living in a fog. It was all a blur. I do remember that I had a really hard time. I just couldn't cope with a demanding baby and still take care of myself. I just felt lost and had nothing left to give. I think that's when it [feeling disconnected with her child] all started. I knew then that I wouldn't be a good mom.*" [61]

**2.2 Striving to be a 'good' or 'perfect' parent.**   This subtheme links to the descriptive subtheme of *hopes and dreams for the future* (subtheme 1.2). The impact of women's trauma history on their **negative self-belief** (discussed in subtheme 7.1) and their perceptions of being able or ready to parent were evident from pre-conception and across the transition to parenthood. Many believed that they would not be a 'good' parent due to their trauma or that their body was 'holding the trauma', contributing to their decision to terminate previous pregnancies [61] or **delay pregnancy** [46, 48, 53, 54, 64]. Parents described a sense of 'unworthiness' which related to a sense of **fear** and **stigma or judgement** (discussed in subthemes 6.1 and 6.2).

"*I never wanted to be a mum . . . it happened anyway. I wasn't going to be a mother because I just didn't think I would be a very good mum. I wasn't very confident.*" [48]

One study [61] included a small number of survivors who were **childless by choice** because of their childhood sexual abuse. These women struggled with intimacy or fertility, yet also felt unable or unworthy of mothering. Their internal beliefs were amplified by other people's perceptions that they were unable to be a 'good' mother or would be at risk of abusing their child given their abuse histories.

"*I believe this is true for so many of us, we believe that there is something about us that made that [abuse] happen and that we are really at high risk. . . Almost everybody could be talked into thinking you can't possibly have children because you will do the same things to your kids that was done to you.*" [61]

Women also shared positive accounts of their changing identity. Becoming pregnant and the decision to keep the pregnancy gave some women a sense of **responsibility or purpose** [49, 53–55, 61, 65].

"*My purpose was to grow a baby and to be a mother so that was a huge shift.*" [61]

As part of their new role, mothers reported a sense of responsibility associated with providing for their baby's growth and emotional needs [49, 61, 65] but also a heightened responsibility to take care of themselves for the sake of their child [53, 61], as previously discussed in subtheme 1.1.

"*If I stay in bed and be depressed all day, then nobody is going to take care of this little girl. I just, you know, I have such a thing against abusers that I have to make sure that it doesn't happen to my little girl. I have had to be a lot more responsible about how I think and about how I feel.*" [53]

Young single mothers also described their **determination** to rise above adversity and "*do what [they] needed to do*" to become the best mothers they could be [55]. Motherhood helped some women to **regain a sense of self-worth** and of belonging in the world [53, 67].

"*What's so outrageous for me is that it's the most normal thing in the world is to have children and a family. Like it's so normal . . . but it's a good normal. You get to understand, 'Oh, that's what my breasts are for', 'That's why I have hips' . . . it's like all of the sexual connotations of women's bodies—the curves and all of that—it's so crude in a way. Because breasts are for breastfeeding, you know? Hips are for birthing a baby*!" [53]

Having felt trapped and fearful of their parents as a child, some incest survivors were acutely aware of their own power over their child and particularly sensitive to their child's needs and emotions [47]. This heightened sense of responsibility weighed heavily on some mothers and links to a **fear of repeating the past** (subtheme 1.3). In contrast, some mothers worried that their own past experiences would hamper their sensitivity as a parent [59].

There was considerable diversity in women's perceptions of **being able or ready to parent** [55, 61, 65, 67], and therefore their parent identity. How prepared women felt as they approached motherhood was influenced by whether the pregnancy was planned or wanted, and whether this was their first or subsequent pregnancy. Their childhood experiences, mental wellbeing and stage of recovery were also significant.

"*I just didn't think that I wanted kids, I didn't know what to do with them, I didn't know if I could love them because I didn't know what that was.*" [65]

"*I had to look at that really carefully, when I found out I was pregnant, when I was thinking about it, you know, can I be responsible for this other person? Can I give them, you know, hopefully give them the life they deserve, or to the best of my ability?*" [67]

Despite the challenges, some parents described conflicting **expectations versus reality** [65] and a vision for a **perfect or idealised family** (subtheme 1.2), based on concepts that they had understood from 'normative discourses' [45]. Their **identity** [46, 53, 54, 57, 61, 64] as a mother or pregnant woman was often positioned within the view of what it meant to be a good pregnant women or **good mother** (subtheme 1.2), and further defined by their abuse history. Parents described distress in **containing trauma and 'being good'** [45, 58, 64], which was rooted in fear of **stigma or judgement** (subtheme 6.2) that if they were not good, they would be viewed as "*mad, bad, hysterical and overly emotional*" [45]. Some women described hiding their 'negative' emotions or being a model patient to "*get through*" pregnancy [64].

"*I think I was conditioned right from an early age, so that programming's always gonna be inside me and when someone's either like threatening my personal like intimate spaces or hurting me, or telling me to do stuff and I'm like feeling threatened, I will do exactly what they say. I will be the best patient they can possibly have. I'll be that star patient. But I'm not. I'm actually screaming inside. I'm like absolutely terrified. I'm expecting them to hurt me. I'm being good because I don't want them to hurt me anymore.*" [58]

**2.3 Wanting to be 'normal'.** Parents spoke about **wanting to be 'normal'** [57, 61, 64] and described challenges with the **identity** of being a 'trauma survivor', which conflicted with an emerging **identity** as a mother and perceptions of **a 'perfect' or idealised family** (subtheme 1.2). *Wanting to be 'normal'* links to parents' explanations for discussing the importance of **containing trauma** (subtheme 2.2), **disclosure of abuse** and **normalising care** (discussed in subthemes 4.1 and 5.3).

"*I hate the idea of sometimes going round as if you've got a label on your head that says 'I've been abused'. And you just think 'I don't want that'. I want people to treat me as normal.*" [57]

"*I don't know what normal is, but you want to be normal . . . like everybody else.*" [64]

**2.4 Knowledge and learning about parenting.** Mothers' described their **lack of knowledge** [53, 62, 64] as key to their perceived parenting ability during this transitional period. Women discussed **learning about pregnancy and parenting**, or what not to do, through reading, television, observing other mothers' interacting with their children, talking to close friends and family, self-analysis, and counselling [53, 54, 65, 71].

"*I felt like I learned how to be anything different than my parents by kind of like reading books, and kind of looking through other peoples' lives in different little ways, like just being there in somebody else's house, seeing a mother or grandmother just doing something that my mother would never do.*" [65]

Some women felt that their knowledge was limited due to their abusive family situation, leaving them completely unprepared and naïve about pregnancy, childbirth and the early postpartum [62, 64]. Expectant mothers in one study sought information about the potential impact of sexual abuse on pregnancy, labour and childbirth, by reading and seeking professional advice [54]. Parents described how the above resources for **learning about pregnancy and parenting**, as well as **parent skill training** [59, 67], **access to information** [53, 74] and **reassurance** from others [64] helped them to develop parenting self-confidence and overcome some of the challenges described and achieve their aspirations of a better life for themselves and their child.

## Theme 3. *Feeling connected*: The quality of relationships with self, baby and others has major impacts on the experiences of becoming a parent

This analytic theme includes descriptive subthemes around: *new experiences of love and joy*; *relationship with self/body*; *relationship with child/bonding*; *relationship with partner (and IPV)*; *relationship with family of origin*; and *other relationships and support*.

**3.1 New experiences of love and joy.** Giving birth and becoming a parent gave many mothers an overwhelming sense of **love or joy** [47, 53, 55, 61, 65, 67, 73] and a "*healthy love*" that they had not experienced before [47, 53]. Some mothers described needing to be loved, or wanting someone to love, as part of the reason they became pregnant, and that their baby's love for them would compensate for their feelings of abandonment, inadequacy and emptiness [47, 61, 67, 73].

"*When I found out I was pregnant I was on cloud nine, I was so happy. I said, I'm gonna have my little girl. I wanted a girl very badly because I wanted to give this little girl all the love and*

*everything that I didn't get. I wanted a little girl for me, so that when she grew up she'd be beautiful and loving, and we would be friends, a relationship I did not have with my mother. So in a way I wanted the little girl to kind of grow up as me.*" [47]

Cohen [47] suggests that this abuse history may make mothers more vulnerable to becoming enmeshed with their children, which can create challenges with **establishing healthy boundaries** (discussed as an important coping strategy in subtheme 7.2). In contrast, one mother firmly stated that her parenting was not shaped by a desire to be loved, but rather by a desire to parent differently.

"*I didn't go into it because I wanted to live vicariously through this little person, which I think to some degree my mother did, and I didn't go into it expecting this is how I want to be loved, this is where I get my love from, and I think that that has made a huge difference in how I parent.*" [65]

Parents also described the **primacy of the baby's needs above all else** [64].

"*I guess in my mind everything was for the well-being of my baby and that's what I was concerned about.*" [64]

**3.2 Relationship with self/body.** As previously discussed, many women in this review reported mixed feelings and experiences during pregnancy, birth and the early postpartum period, often related to their experiences of sexual abuse. Many women had negative perceptions of their **body in pregnancy and birth** [46, 51, 53–55, 61, 64, 70], and how their bodies were profoundly changed because of having a child. Some mothers described feeling that their body was damaged, inadequate, defective, untrustworthy or simply bad, which linked to feelings of **shame** and experiences of **triggers** (see subtheme 7.1). Conversely, other women described a perception of their **body in pregnancy and birth** as powerful, capable and strong [46, 47, 53, 54, 57, 61]. These experiences related to a sense of *empowerment, choice and control* (subtheme 5.1).

"*When I was pregnant I felt very proud, real proud of having a baby, I just enjoyed the whole thing. It made me feel like I am a woman. Like powerful. Yes. I really felt powerful.*" [47]

"*Physically having a child was a really profound experience for connecting with my inner child, it was profound. I thought I wasn't prepared for that or I didn't understand the magnitude until you actually give birth and you see this child and that sense of being overwhelmed. This it also helped me to honor that inner child and imagine what I was like as an infant type thing so it helped melt it together and you know again the innate wisdom that babies have like how smart they are about what they need and I was being able to live with a child who knows innately what they need and this has taught me so much, how they survive and how they thrive. I feel like I have finally made it. I feel safe. I feel balanced.*" [61]

Some negative experiences of their body during pregnancy and birth were specifically related to trauma [46]. Negative body experiences also included **co-morbidity or illnesses in pregnancy or physical symptoms** [54, 55, 70]. Parents in several studies described the **baby as feeling foreign and invading their body** [53, 57, 64], which were related to feeling a lack of control.

"*When I thought about the baby being inside me, I would just get sick. Like I had to come to terms in myself and allow that thing to be in there, like giving permission. It sounds really*

*weird and convoluted, but that was something that I went through very strongly and I think that the reason I knew that they were related is that I had so many nightmares during those times . . . I knew inside me that the baby was a baby and mine, but just where it was habituating was the issue for me. Just being in that private area.*" [64]

Some women questioned whether they could become pregnant because of their **sexual traumatisation** [46, 51] and pain or reproductive dysfunction [53, 61]. Some women with a history of sexual abuse also spoke about their difficulties in having sexual intercourse and a fear of intimacy [46, 51, 54, 61].

"*I found it hard to sleep with him to get pregnant I found that quite um traumatic and the thoughts of it to start with I was sort of going through how I could get pregnant without having to have sex . . . I sort of forced myself in the end thinking that's this is the only way.*" [46]

Some women who had been struggling with **fertility issues** [53, 54, 61, 65, 67] were surprised they had conceived. Some considered their bodies as "*spoiled*" by the abuse and reported a **conflict between being 'spoiled' versus wanting to be 'pure'** [53, 67]. This links to concepts about **a 'perfect' or idealised family** (subtheme 1.2).

"*. . .and I never did once feel uncomfortable, but then I look back and I think, did I block it because I wanted this to be a good experience? And, I loved being a mom and I loved being pregnant. Maybe I didn't allow myself to feel negative thoughts, I just wonder now that I look back if I just didn't want anything to spoil it 'cause I had felt like my life up to that point had been spoiled, tarnished, and I didn't want anything to (voice breaks) harm my memory.*" [64]

**3.3 Relationship with child and bonding.** Parents described mixed and positive experiences of **bonding** [61, 62, 65] as well as challenges with **early attachment** [51, 53, 60–62, 64, 65] with their child. This relates to experiences of postnatal depression, **negative self-belief**, perceptions of a '**perfect' or idealised family** and **fear of repeating the past** (see subthemes 7.1, 1.2 and 1.3, respectively), which also link to concerns about **baby gender** (outlined below).

"*The first thing I felt when they put him on my stomach, I thought to myself, I feel clean, you know like all that bad stuff was washed away.*" [62]

"*When my son was born I was not able to bond . . . I just pushed him over to his father. I didn't want anything to do with him, but I wanted to be sure that he was safe and protected, in good hands . . . but it needed to be somebody else . . . not me. Because if it was me, then he . . . he could possibly be a pedophile, he could be like my father. That is why I told myself through my pregnancy that I was carrying a girl. There was no room in my mind for the baby to be a boy, and so when James came out, he had no name . . . My husband had to name him because I . . . I couldn't even go there.*" [53]

Parents described positive experiences [46, 49] and challenges [46, 47, 49, 51, 53, 57, 61, 65, 74] of **breastfeeding**. Survivors of child sexual abuse also discussed positive experiences [49, 61] in relation to **bonding or attachment through breastfeeding,** however some parents noted intense feelings of discomfort [49] and **dissociation during breastfeeding** [74].

"*It's the love. It's the giving of my milk to him and sharing with him. I am the only one that can do that for him and it is so strong that love. To have this little baby attached to you makes me feel a really strong connection.*" [49]

"...it's like it's happening again because you are being controlled by another person. And even though I really tried not to feel like that, it happened every single time um that I tried breast feeding. I felt that immense feeling of being controlled by someone else." [58]

Some felt their ability to breastfeed reflected on their ability as a parent:

"She wouldn't breastfeed, it was all what am I doing wrong, what is so wrong with me, why can't she latch on, why can't she attach to me, and that was a pattern that was repeated that was my mother and my dynamic." [65]

For some parents, the **baby gender** [47, 68] created fears for the child, which impacted on the **parent-child relationship** [47, 70].

"I worried a lot. I've always been so anti-abortion, but when I found out it was going to be a girl, I wanted to give her away or just not have her. I was so scared I wouldn't be able to protect her and that something would happen to her. It made it hard. It made it hard a lot, because I thought about [the sexual trauma] a lot when I was pregnant." [68]

**3.4 Relationship with partner (including IPV).** Mothers described a **positive relationship with their partner** [53, 67, 68, 71, 73] and considered **partner support** [54, 67] as critical during the parenting transition. However, **challenging partner relationships** were also frequently reported [45, 53, 55, 60, 61, 65, 67, 68, 73].

"I met my baby's dad and we knew instantly that we loved each other and that we wanted to be with each other. We talked about what we wanted out of life and decided that we both wanted to be parents. I took into account his family and how his parents are . . . how his family lived . . . I thought, 'You are going to be a good dad'. When I decided to have a baby with him, it was me wanting to share my love with him. It was my way of saying, 'I love you and I know that you are a good person'. It was a whole new beginning." [53]

"Like he knows what I've been through, but it's just confusing for him because, 'Why are you acting like this now?' And like I said, he's already sceptical about this baby. And now I'm pushing him away more. So our relationship is going through hell in a hand basket." [45]

Many women described break-down of relationships during pregnancy, which may impact on some mothers who wished their child would look like them [55]. Several women described experiences of **IPV in pregnancy**, which can compound the complex trauma experience [56, 69]. However not all women wanted help from maternity care providers to address IPV [69].

"He [the father of the baby] heard that I was pregnant and wanted to beat me up. He beat me so that I would lose the child but I didn't lose it." [56]

One parent also reported challenges with their **partners' family** [55].

**3.5 Relationship with family of origin.** Parents described mixed experiences in their relationships with their **family of origin** [47, 54–56, 61, 65, 67, 73] during pregnancy, birth and early postpartum. For some, becoming pregnant was a time for a 'fresh start' with this relationship, setting new boundaries and 'letting go' of the past [54], receiving practical and emotional support [55, 56, 67], and having an opportunity to reconnect and heal [67, 73].

"*Everything is changed since I've had a baby. For my family, now they all have to treat me as a responsible person—my entire family seeing me become a mother definitely has changed who they think I am in their minds. That's the biggest way my relationships have changed; they have to re-evaluate who they think I am, and my relationship with them changes accordingly, because I haven't really changed that much, I'm still the same person I was before I got pregnant and while I was pregnant, except I've had to start being more healthy, but essentially I'm the same person, just minus and plus a few things. But motherhood has meant a major change of status in their eyes; I had 'irresponsible drug addict delinquent status', and I cannot be that anymore.*" [67]

Other parents described challenges in their relationships with their **family of origin** [47, 54, 61, 65, 67], including learning to trust [54] and ongoing conflict [67]. Some of these challenges seemed to be associated specifically with adolescent pregnancy [67]. Forming a maternal identity was associated with internal conflicts as women set new boundaries with their families and with their child [46, 53, 54, 61].

"*And they know, they are very clear, both of them, that they are not going to mess with me ever again. That there are definitely boundaries. There are definitely things that I'm willing to accept from them and things that I'm not willing to accept. And that behavior, I am not willing to accept.*" [54]

**3.6 Other relationships and support.** Parents described the importance of **supportive relationships** [53, 55, 61] and role models during pregnancy and early postpartum as well as having **support people during birth** [55, 73]. The **need for support** [60, 61, 64] was highlighted as an important factor in healing during this transition, with the types of support detailed in theme 7.

"*He [husband] was beside me every step of the way . . . even when I pretended that I didn't need him, he stayed with me. I don't think I could have done it without him.*" [61]

"*I had a really special girlfriend that I could share everything with . . . so she was a life-saver when it came to my pregnancy. I don't know how many times I called her crying my eyes out . . . she never judged me or said I was bad, she just came.*" [61]

However, a **lack of support during pregnancy** [52, 53, 60, 61, 68] and **in early postpartum** [61, 62] was a challenge frequently highlighted, with some parents describing intense feelings of isolation.

"*I didn't have a lot of friends and I didn't have a lot of support . . . I was alone with my son and I was sick. That was really tough; those nine months were really, really hard for me.*" [53]

This was particularly important for parents who may have had difficulty developing supportive relationships with others, including **friends** [61]. It raised conflicts and challenges for some parents, between their need for support and the instinct to push people away, or the experience of having trouble **letting others in** (see subtheme 4.1).

## Theme 4. *Compassionate care*: Kindness, empathy and sensitivity enables parents to build trust and feel valued and cared for

This analytic theme incorporated descriptive subthemes of: *provider support, communication and relationships*; *trauma-informed care and factors which factors which foster safety and enable care*; and *experiences of care during birth*.

**4.1 Provider support, communication and relationships.** Parents described positive **perinatal care experiences** [56], with **supportive providers** [51, 53, 59] as key to influencing their experience of perinatal care.

> "*People who believe them. People who validate [what] they're experiencing; who don't compare them to other people who . . . help them take responsibility for their role but . . . don't blame them for it; I think like, ya know, try to relate to your population.*" [59]

Some parents referred to care during pregnancy and birth as being '**normalising**' [46], which links to *wanting to be 'normal'* (subtheme 2.3).

> "*It was very different to become this normal person going to normal hospital appointments um that normal people would do I know this is all very normal but um, it was it was quite a nice experience thinking I'm doing things that um people will do without mental health problems.*" [46]

However, parents described how it was difficult to **let others in** [53] and **trust in care providers** [48, 51, 53, 57, 62, 64, 69], which links to a perception that the *world is unsafe* (subtheme 6.1). Parents also described many challenges related to a **lack of care, empathy or understanding** [51, 53, 57], **inadequate care** [53], and **poor provider communication** [46, 51, 53, 61, 64].

> "*I was exhausted after the birth . . . I said, 'I feel ever so weepy'. 'Oh, don't start that off!' she said, 'I'm not going to be 5 minutes!' I mean, she was a cow. She really was awful!*" [51]

These negative experiences related to parents' feelings of **being disregarded or depersonalised** [48, 51, 53, 57, 61, 64]. Their experiences reflect the importance of 'compassionate care', with women describing concerning experiences of lack of care and respect for their choices, particularly in light of trauma.

> "*. . .it was when doctors sutured you up . . . it was an SHO [senior house officer] who'd obviously been dragged out of bed. . . He didn't look at me once, didn't, didn't sort of get eye contact whatsoever . . . and I felt every single stitch he put in, every single, and I cried all the way through.*" [51]

> "*I didn't want to be strapped down. I didn't want anything in my arms during my first pregnancy. I didn't want to be on a table. I wanted to walk around. And I had this nurse from hell. She was just so tied into what you have to do and what the rules are. And, oh, she got me in that bed and she just put those needles on . . . in me and you're gonna be in bed. And I just stared to panic all over again. And I kept telling her I want, I just want, I don't want the monitors on me. I want to be up. I want to be able to get up to go to the bathroom and then they put the catheter . . . and you're there . . . it's like being a prisoner all over again.*" [64]

Often, these experiences led to preconception, pregnancy and childbirth **care being reminiscent of abuse** [46, 48, 57, 58] with women **re-experiencing or being triggered in care settings** [48, 51, 53, 57, 58, 61, 66, 74] (links to subtheme 7.1). Some of this relates to shared experiences of severe perineal and abdominal pain, constraint and a lack of control during labour and birth. Some women described their care as **invasive and violating** [48, 51, 53, 57, 61, 62].

"...with him [stepfather] there was no care for who you were . . . He always reckoned he loved you but there was no care for who you were, which is why when there's like the midwives and the doctors that are just 'Oh, I've got to do my job' sort of attitude—it's that 'no care' the same as what he gave'." [51]

"I think I was angry about how I was treated at the hospital and it took me awhile to think through why I felt so uncomfortable in the hospital. It wasn't until Emma was four or five months old that I finally realized, 'They had no right to touch my body like that! They had no right to treat me like that!'" [53]

Coles [48] argues that, due to the challenges of *disclosure of abuse* (subtheme 5.3), perinatal care providers should adopt "universal precautions" and provide sensitive trauma-informed care for all mothers.

**Effective communication in the care setting** was identified by many parents in this review as critical to improving their perinatal care experience [48, 51, 54, 59, 66, 69, 72]. Communication was also critical for enabling safety and the *disclosure of abuse* (subtheme 5.3) [72]. Some parents provided examples where they were very directive about the care they needed during pregnancy and birth, but this required a fair amount of individual agency. Positive descriptions of care involved **respect, empathy and understanding** [51, 61], and **personalised individualised care** [48, 51, 64]. Positive experiences of providers advocating on behalf of parents were also described [54].

"I think the nature of the person, like the nurse I saw last week before my scan. She was talking to me and asking me how did I like living here and interesting herself in me as a person before she did any of the examination and explaining what she was going to do and why she did it. So, I suppose, being acknowledged that I'm a person there, rather than an object on a conveyor belt of vaginas that she's looking at." [51]

Parents described the importance of asking people for permission before touching them, being gentle and engaging with people as individuals, with compassion and understanding [51].

"... and she [midwife] said, 'may I examine you?' And I let her examine me as well . . . She asked my permission first and said, 'this is what I can do' and I gave her my permission, and she went ahead and did that. She was very, very gentle, she was lovely." [51]

"Like you could just feel like her attitude was just common sense, down to earth, trust your body and also I'm not going to do anything that you don't want me to do, you tell me about how this has happened, where do you feel it and you could just feel that the way she was talking to me she was just totally respectful and she was like you are the expert here is some ways . . . you know, that kind of feeling. She made me feel safe . . . she made me feel like I'm not stupid here, she's not just the expert here telling me don't do this, do that, that kind of thing . . . but this is your body and I know that we are talking about your body and that the kind of thing that was great. Yeah, this is my body and it has been traumatized." [61]

**Lack of continuity of care** and needing to repeat their story was noted as distressing by some parents [59]. **Continuity of care** [48, 50, 51, 61, 62, 66] was emphasised as an effective strategy for minimising these issues and building relationships and trust with providers, which link to subtheme 6.1.

**4.2 Trauma-informed care and factors which foster safety and enable care.** Parents discussed other challenges in relation to **postpartum access to care** [48], which relate to **stigma**

(subtheme 6.2), **avoidance of care** (subtheme 7.1) and **others' expectations** [61] as reflected by the following:

> "*Basically I just didn't go to see a doctor or take prenatal classes or anything like that. I really didn't want to get all caught up in the pregnancy thing . . . I had enough going on already and I didn't want to have to be all nice and cheery and tell everyone that everything was wonderful, because it wasn't wonderful. I hated being pregnant but that's just not something you say. People would think that I was a bad mom and I already felt that inside . . . I didn't need to feel it from others.*" [61]

However, parents described many positive experiences and examples of **'trauma-informed' perinatal care** [46, 48, 53, 54, 57, 59, 61, 64, 69, 74].

> "*. . .and by this time I could tell I was teetering on the brink of having everything rush back. And I was visibly shaking, my voice was very shaky, I sounded like a little kid when I talked. And she, the midwife, really picked right up on it, and she said before she examined me. . .she sat down and she explained exactly what was going on. . .she said the same thing six different ways until I physically. . .you could physically watch me relax. And then she said, 'Why don't we just take a peek and see what's going on?'*" [69]

Key elements that helped with fostering positive relationships and safety to counteract trauma-related shame and mistrust in others included **positive strengths-based approaches** [59], **continuity of care**, **multi-disciplinary care and collaboration**, including family-centred care [59] and **improving professional contact opportunities** [48] to ensure the frequent contacts with providers during the perinatal period are used to their full potential. Muzik et al [59] reported that many mothers discussed the need to emphasise hope and healing as opposed to trauma which implies injury and damage, and argued for the need for "hope affirming practices" and child-friendly services that support parents to achieve their hopes and dreams for the future with their new child.

Two studies outlined a comprehensive range of factors to be included in trauma-informed perinatal care. Muzik presented survey findings to suggest that "many mothers (94%) welcomed the idea of access to a range of multi-disciplinary, holistic, healing and well-being services that meet their needs as trauma survivors, as women, and, particularly as parents" and outlined 10 important aspects for these services [59]. Seng presented a similar set of guidelines for providing trauma-informed perinatal care, which were considered relevant for all women at different stages of the healing continuum [69].

**4.3 Experiences of care during birth and breastfeeding.** Parents described mixed positive experiences [51, 53–55] and challenges [46, 53, 55, 62] related to **birth**.

> "*They had to tell me to let her go so that they could weigh her and everything. They was like, 'You've been holding her for an hour now'. You know, but my first reaction was to put her on my chest . . . I was just like, 'My baby!' It was like, I don't know, it was like a dream . . . I was just so happy to see her face.*" [55]

> "*It was like I didn't like the feeling of labor or any of that . . . I was like, staying in the hospital and all that, ugh I hated it!*" [55]

Some women described the **healing process of birth** [54, 61] as a positive experience, with birth seen as a transformative experience and a "turning point in their perceptions of their abilities and the possibilities for their lives" [54].

"*It was just the most wonderful, magical experience of my whole life. I feel like things changed for me right there.*" [61]

**Support people** [55, 73] (discussed in subtheme 3.6) were essential elements of a positive birth experience. However, some mothers described challenges with having a **birth partner** [51, 53, 62, 64], and feeling alone. These challenges included restricted visiting hours [51], partners being absent and family not being supportive [55].

"*They were telling me what to do, and what I couldn't do and . . . I can remember saying that I felt you know, the pains were really bad because I went to the hospital and they sent my husband home because I was obviously in early labour and um . . . and I was completely on my own through the night, wandering around corridors . . . trying to keep quiet because people were trying to sleep.*" [51]

Positive birth experiences were also associated with positive feelings about the **body during birth**, **bonding**, and **empowerment or control** (subthemes 3.2 and 5.1). Several women talked about their **hopes for the subsequent birth** [53, 54], reflecting on previous experiences, and aiming for greater control during the process. Parents also experienced challenges during birth related to **traumatic birth experiences** [53, 55, 61], **stillbirth** [53] and **pregnancy loss** [53].

Many women also described **breastfeeding challenges in the care setting** [48, 50, 51, 53, 61, 74], particularly those who had experienced previous sexual abuse.

"*When I was breastfeeding, I was all exposed but they [nurses] didn't seem to care. They were like, 'we've got to get this baby to eat so we're going to do whatever we have to, we need to make sure the baby eats.' And they didn't seem to care about the fact that they were pushing my body around.*" [61]

## Theme 5. *Empowerment*: Control, choice and 'having a voice' are critical to fostering safety

This analytic theme incorporates descriptive subthemes of: *empowerment, choice and control; having a voice;* and *disclosure of abuse history.*

**5.1 Empowerment, control and choice.** **Lack of control** and feeling overwhelmed during pregnancy and birth was a strong theme discussed by child abuse survivors [46, 48, 51, 53, 57, 60–62, 64, 70]. This included women feeling like they were passive rather than active participants in the birth process, the lack of privacy and the 'medicalisation' of birth.

"*I knew I was going to have to deliver this baby . . . and I knew there were going to be more examinations and things being taken out of my control again, because I didn't feel strong enough to say 'No, I don't I want you to do this', or explain the reasons why I would be behaving in certain ways.*" [51]

"*I felt like I had no choice, it's like whomever came in had a right to touch me. It's a pretty vulnerable place to be in.*" [61]

These feelings of lack of control were linked to feelings of **being disregarded or depersonalised** (subtheme 4.1), **vulnerability** [57, 64], **shame/humiliation** [51] and fear (discussed in theme 6). Some parents talked about a sense of inadequacy and childhood experiences of being abused and sometimes not being believed by people in positions of **authority** [47, 48, 51, 53, 57, 61, 62, 74]. This impacted on their current experiences of perinatal care in feeling

vulnerable, developing **trust in care providers**, and **care being reminiscent of earlier abuse**. **Care provider gender** was also highlighted as impacting on relationships with care providers and a sense of control. While most mothers said they felt less vulnerable if the care provider was female [51, 57, 59, 61, 64, 66, 69], this was not universal with one women expressing a sense of relief at "*not having to explain anything to a male*". [64]

Unsurprisingly, parents emphasised **empowerment, choice and control** as a critical element of positive birth experiences [53, 54, 61, 62], and for helping to improve perinatal care experiences [51, 54, 57, 60–64, 66, 69].

> "*Labor was probably my biggest success. I was present all of the time, natural, complete, and supported . . . certainly aware of the pain and the changes and all of that, but it was very empowering because of that 'hey, I can do this' feeling. It is maybe the first thing that I can do completely, be in charge of and that really was a drive for me. It turned out to be positive that I could focus so clearly on that. I guess my fear before was that I would feel overwhelmed that I would feel violated, that I would feel this was being done to me and out of control. Those were my fears around giving birth so by actively working around those issues, that I could turn them around and that I was able to do that. I could feel that I was being violated somehow but surrounding myself with a support network and realizing that they were all there honoring what I was doing it just helped.*" [61]

Some parents described how **having a birth plan** [54, 61] or **home birth** [51] helped them to achieve more control. One parent described developing a birth plan with a trauma specialist [61].

> "*I think the birth plan made it really positive and we discussed it with my doctor . . . about certain issues in the birth plan and then I would go over it myself and then talk with my mom and then during the prenatal classes . . . I loved my prenatal class, I thought it was great. I talked about it at prenatal class and this class was taught by a doula, which was great.*" [61]

> "*I didn't want to be part of the conveyor belt system, so I decided I was going to have a home birth.*" [51]

**5.2 Having a voice.** Parents described challenges with **talking about trauma** [54, 57] **and birth experiences** [61] and the anger, frustration, resentment and suppressed feelings that could build from not talking.

> "*I laughed when visitors came and I smiled and I put the right face on. But inside. . . (sniffs, four second silence). Inside I was–just silently screaming.*" [57]

**Asking about trauma** [53, 69, 72] (which links to *disclosure of abuse*, subtheme 5.3), **expressing trauma** [53, 54, 61, 68] and **communication** [65] were all seen as helpful by parents. However, there was no 'one size fits all' approach and there was a lot of variation in both the readiness and timing of parents for 'telling their story', and the way they wanted to do this [54].

> "*I guess not keeping it bottled up inside of me probably made, played a major part in my life because I felt that whenever I did keep it bottled up, I felt like I'd went through a lot; and now that I've talked, I'm more open about talking about it, I feel it has changed my personality a lot. And, then like, I don't feel that pressure there anymore as much as I did before when I didn't say anything . . . I kept it bottled up for a long time.*" [68]

White [72] outlined five themes for safe trauma enquiry: "(1) a clear definition of trauma, (2) clear purpose for inquiry, (3) reassurance that inquiry was routine, (4) confidentiality, and (5) mention of helpful resources other than psychiatric therapy."

**5.3 Disclosure of abuse history.** **Disclosure of abuse** was perceived as positive in terms of enabling support and more sensitive care, as well as advocacy for women [54, 61]. But there were also many challenges in the **disclosure of abuse** [46, 51, 53, 54, 57, 61, 64, 69], including not being asked [46] and the **impact on family** [51], with a high degree of variability between parents. This relates to previous descriptions of women feeling they are invisible and silent while 'screaming inside'. This silence creates challenges for perinatal care providers and could lead to parents feeling they are being disregarded and overlooked. Hence recommendations for "universal precautions" [59], as previously discussed (subtheme 4.1), in the context of parents' childhood experiences of maltreatment are salient.

"*Pregnancy is a time for purity. I want my pregnancy to be pure and not tainted with memories of a dirty past . . . even though it wasn't my fault. I don't want to be talking about horrible stuff like what happened to me as a child and I certainly don't want to feel like I am going to do the same to my child. I just don't think pregnancy is a good time to talk about it. I just didn't even want to think about it then.*" [61]

These challenges were related to **fear** and **trust in care providers** (discussed in subtheme 6.1), **fear of losing the child** [46, 57] and **fear of not being believed** [46].

"*I never came forward because I thought they were going to take my baby away'.*" [46]

"*Unless we can take away this awful cliché that the abused become the abusers. Then, I don't know if anybody will ever be really free of fear enough to talk . . . [I was] terrified that at some point, somebody was gonna find out, that I'd been abused and that they were going to put this label on me and watch me. And look out for the signs that I was somehow dodgy.*" [57]

Other challenges parents raised in disclosing abuse included **not wanting to burden others** [providers] with abuse stories [57], **wanting to appear normal** to providers [57, 61] and to minimise the impact on themselves (as discussed in *wanting to be 'normal'*, subtheme 2.3).

## Theme 6. *Creating safety*: Parents perceive the 'world as unsafe' and use conscious strategies to build safe places and relationships to protect themselves and their baby

This analytic theme incorporates descriptive subthemes of: *the world is unsafe and strategies to protect themselves and their baby*; and *the external world around them*.

**6.1 The world is unsafe and strategies to protect themselves and their baby.** Concerns about safety were a major challenge raised by parents in this review. Challenges were raised in discussions around **safety in disclosing their abuse history** [64, 69] and **safety or protecting themselves and their baby** [45, 47, 48, 54, 61, 64, 68]. These perceptions of the world as unsafe could lead to parents being overprotective of their child in **monitoring safety and hypervigilance** [48, 53, 64] and encountering challenges in **establishing healthy boundaries** (subtheme 7.2).

"*I realize that she is going to hurt. But God forbid that she should go through what I went through. I'm hoping that if I keep an eye on her, and watch the people that are around her,*

*that that will be enough. I'm hoping that if I do what I'm supposed to do, that God will protect her from that. She can have other horrors, but just not, not that.*" [45]

"*Just more watching people change her, being more curious, and just wondering what people are thinking when they were changing her and what are they going to do—just not letting them change her away from me either—I like to hover.*" [48]

For some, protecting their baby involved managing relationships with their **family of origin** (subtheme 3.5) [54]. Some women even gave their children up for adoption in an effort to protect them from their own family:

"*The baby was a girl and I knew that I couldn't take that baby home. . . because she was a girl, because she wouldn't be safe because my daddy lived at home and he was on to my sister at this time. I just knew that I couldn't bring that baby home . . . Most of us women gave up our kids for adoption . . . there was no support in the family. . .*" [61]

These concerns about safety were strongly related to challenges in **trusting others** [47, 51, 68], including **trust in care providers** and the need for **empowerment, choice and control** (subthemes 4.1 and 5.1).

"*I was scared of other people. . . many people thought I was shy, but it really was not that I was shy; I just did not trust anyone, and I really did not want to talk to anyone.*" [68]

**Creating an environment of trust** [64, 65] during the transition to parenthood and **building trusting relationships with care providers** [59, 66] were seen as important aspects of improving perinatal care and building relationships with providers.

"*I've been sexually abused, understand there are boundaries with a person like that . . . besides for warning me, let me know, 'Okay, I understand that you've been through this situation, I'm gonna do this, I'm not trying to make it uncomfortable for you.' Let the person like me, in my situation, know you're not out to hurt me. That way I can gain that trust from that person and know that okay, it's gonna be okay. And with each visit, not just sometimes, with each visit.*" [66]

**Regaining a sense of safety** [46, 61] was described as a positive aspect of pregnancy and birth by some parents.

"*I felt like nobody would touch me because she is a pregnant woman—that kind of thing. Like nobody would attack you . . . you are treated with more kid gloves. I felt safe somehow and special'.*" [61]

Parents also described a strong positive commitment to ensure **child safety and protection** [45, 53, 59, 65], which relates to a **fear of repeating the past** (subtheme 1.3) and could lead to overprotection and challenges setting healthy boundaries.

"*[I want to learn] how to parent with healthy fear because I think that anyone who has had one of those traumatic events is going to have fear of it happening to their child and there has to be a line between protecting your child and overprotecting . . . learning how to let them live but still protect them and teach them.*" [59]

**Safety in perinatal care** [48, 62, 64] was identified as critical, and is strongly related to *empowerment, control and choice* in care (subtheme 5.1) [64]. Parents also emphasised the importance of clinical **environments that foster safety** [59].

"*I think a place that's nice and kind of cozy . . . that doesn't necessarily feel like a doctor's office or a hospital so that you feel like you want to be there. I think that's also important so that you might choose to go rather than sort of hide away . . . I think that's important . . . a comfy atmosphere so that you really feel safe'.*" [59]

Richmond (2006) proposed mechanisms which demonstrated the link between showing regard for a person's individuality and needs, and communicating respectfully and offering choice and control–which helped to foster trust and a sense of safety in perinatal care [64].

"*I appreciated that he always would tell me what he was going to do before he touched me or did anything and that helped a lot, even though it was very difficult.*" [64]

Some parents chose isolation and care options like **home birth** (subtheme 5.1) to foster choice and control and a sense of safety.

"*My isolated situation in which I was living was protecting myself from the outside world because I knew that I could cope in a situation where I'm isolated because I've only got me to deal with then and I could sort of hide, I could go bush, I could do what I had to within my own isolation.*" [62]

**Protecting the child or ensuring safety** was also seen as an important enabler for healing [53, 65] and relates to *wanting to parent differently* (subtheme 1.3), and achieving an identity that did not resemble the identity of the perpetrator parent [59]. This included developing strategies to protect their child from themselves, which relates to a **fear of repeating the past** (subtheme 1.3):

"*I wanted to protect my kids even if it was from me*" [65]

**6.2 The external world around them.**   Parents described many challenges in their **external environment** [67] that are likely to impact on their ongoing sense of **safety**, and highlight the need for provision of external support for this population group. Challenges include **housing instability** [55, 56, 67] and **financial challenges** [53, 67, 71], particularly for adolescent mothers.

"*When I find out that I was pregnant it came as a shock because I didn't have anywhere to live, no place to stay. It was hard. No one wants a pregnant woman in their house so the time came when I had no choice and I went to the children protection services. I looked for my mum but didn't find her. She had moved, that's why I couldn't find her. I went to the Children's Protection Services and they sent me to the care institution.*" [56]

Perceptions of **stigma or judgement** [51, 54, 55, 57, 61, 65, 67], including **racism and discrimination** [67], were also described as an issue particularly affecting adolescent mothers. This impacted on challenges with *disclosure of abuse history* (subtheme 5.3) as well as choices and experiences of **becoming pregnant** (subtheme 2.1).

"*I could totally tell she was expecting me to be some like flaky, dumb, teenage mom and pawn the kid off on grandma, which you know, so many people do.*" [67]

"*I suddenly felt, 'I can't go outside, everybody knows about me, they're all talking about me'. If I saw people grouped together they were talking about me. They knew it! But I didn't know what they knew, but I knew they knew it. I was very aware in the early stages that there was no way I would look at anybody. It was like . . . just couldn't handle that because I thought that they could see inside to what was going on in my mind.*" [51]

Parents described mixed experiences in relation to **education and employment**, with parents seeing parenthood as a chance for a fresh start [53, 67] yet also challenging due to a lack of education or employment [55, 67]. Again, this was particularly relevant for adolescent mothers who were at a critical education stage and now juggling two important life transitions simultaneously.

"*In the beginning I was working. And then I ended up getting laid, well not laid off, my hours got cut, so I just told them I was gonna quit and started looking into schools . . . I was working at the time, but I couldn't work in the food industry because it made me nauseated so I looked into going to school. . . So I registered into school a few months later and that's about it.*" [55]

Reflecting some of these challenges in the external environment, parents noted the importance of **practical support** [67] during the perinatal period and described a range of strategies they use to access food and support. Some adolescent mothers described **foster care** [56, 67] as a place of safety and source of security, with access to regular meals, shelter, transport and bathing facilities.

"*The shelter is good. You have a place where you can stay which is better than being on the road (sic).*" [56]

### Theme 7. *'Reweaving' a future*: Managing distress and healing while becoming a parent is a personal ongoing and complex process requiring strength, hope and support

Transition to parenthood marks a time in a survivor's life when they themselves are increasingly vulnerable (distress in pregnancy, fear, disempowered in care) and new fears for the safety of their child are exposed. At the same time, pregnancy is often a catalyst for seeking help and an opportunity for healing, to moving forward and making sense, and to "reweaving" a future as described by Pitre [75]. This analytic theme incorporates four descriptive subthemes of: *Distress symptoms, including fear and lack of trust*; *coping strategies*; *factors that help recovery, growth and healing*; and *the healing journey though pregnancy, birth and parenting*.

**7.1 Managing distress symptoms, including fear and lack of trust.** While some parents described **symptom reduction** [54] during pregnancy, most parents described a wide range of **distress symptoms** in pregnancy, birth or early postpartum [45, 51, 53–55, 57, 60–63, 67, 68, 70]. Mothers who had experienced child sexual abuse described particularly distressing symptoms during birth, which midwives perceived were impacting on the process of birth [63]. Distressing symptoms were grounded in **fear during pregnancy** [51, 53, 55, 61], including **fear in care** [51, 53, 57, 58, 61], and reflected survival and self-protective responses utilised to deal with abuse as a vulnerable child. These fears were related to **fear of repeating the past** (subtheme 1.3), **fear of not being believed**, and **fear of losing the child** (subtheme 5.3).

"*I was very frightened. I was very, very frightened and the thought of going through childbirth terrified me. The thought of having people examining me terrified me. Nobody asks you whether it's all right.*" [51]

"*Cos all they do is come in, like mess, fiddle with you, do things to you and then they don't really tell you what they're doing. . .and then they disappear again and leave you and it's almost like you're sort of waiting for the door to open. You don't know who is gonna come through, what's gonna happen, 'n it is very, very frightening*" [58]

Parents described specific distress symptoms including **shame or humiliation in care** [51, 61], **negative self-belief** [51, 59, 61, 64, 67, 68] and **guilt** [48, 57, 64], with some women apologising to their provider for their body and for normal physiological function during pregnancy and birth.

"*I felt like a piece of meat and I felt just like I had when I was being abuse.*" [51]

"*I didn't want the doctor down there [perineum] looking at me . . . touching me . . . I felt so dirty just spread out like that for everyone to see . . . I didn't want them to have anything to do with my body but they kept touching me and telling me to just breathe through it. I puked right then and there . . . just like I used to do when he was touching me. I was so ashamed. I tried to tell them but they just didn't listen. I felt so vulnerable and I had no one to turn to.*" [61]

**Anxiety and depression** [53, 61, 67], including **postpartum depression** impacting on bonding and child development [47, 60, 61] were frequently reported by parents in this review.

Parents described **avoidance or denial** [51, 61, 64, 69, 70, 73], **avoidance of care** [61, 64], and **numbing** [51, 64] as challenges, but also used avoidance as a *coping strategy* (subtheme 7.2) and impacting on **postpartum access to care** (subtheme 4.2). Seng [69] described this "knowing and not knowing at the same time" as the first of three stages in the healing continuum.

Women frequently reported **re-experiencing or triggers during care** (sub-theme 4.1) and **re-experiencing** [47, 53, 54, 61–64, 66, 68, 70], **triggers** [51, 61, 62], **nightmares** [61, 64], **intrusive thoughts or nightmares** [51, 53, 54], or **delayed memory** [57, 62, 63] during pregnancy, birth and breastfeeding. Women described these experiences occurring in a number of situations, not just during intimate procedures such as vaginal examinations. The 'trigger' was an individual, personal experience for each woman. These experiences were associated with feelings of rage and anger, and linked to not wanting to repeat the past, and moving towards forgiveness and acceptance, as reality is confronted.

"*I was scared. I could hear other women screaming, obviously they were screaming because they were labouring too. But I didn't scream, I just swallowed all the sobs and cries because that was the way . . . I . . . did, as a child, swallowed all the sobs, the cries, when I was being abused. I was afraid, I was in pain, um . . . I had a mask over my face and my husband kept trying to put it on to my face which was again, you know, hands over your mouth, when you were being abused as a child to stop you shouting for help. So the whole experience was like being thrust back as an adult but still feeling like that helpless child in the dark and being so afraid and alone.*" [51]

"*I felt like I was stuck, like . . . they made me be there, and that's what happened in the past, that person [the abuser] was on top of me making me be there. It wasn't voluntary, at the hospital it felt like the same thing [the rape] was happening all over again.*" [66]

"*I had those flashbacks or memories all the time. It was like daily struggle to stay on top. There wasn't five minutes a day when I didn't think about it, and having my baby daughter was comforting. And now it's less, but I still think about it.*" [47]

Parents described **dissociation** [51, 54, 61, 62, 69, 70, 74] and **passivity** [64] as a challenge. However, a number of parents also described using dissociation as a 'coping strategy' during intimate perinatal care procedures or experiences.

"*I can do it at the switch of a . . . press of a button . . . I can go off and just not be aware at all . . . which is quite useful sometimes . . . but at the same time you don't hear what people are saying and you don't really take anything in because you're not really there.*" [51]

Many parents described challenges with **substance use in pregnancy** [53, 56, 66, 67, 69, 70] to try to help manage distressing symptoms. However, parents also described an internal conflict and concern for the impact on their baby, as well as the impact on pain management in labour for parents recovering from addiction.

"*I am a recovering alcoholic, so I didn't want to take any [narcotics or sedatives]. . . At that point I was five years sober, so I didn't want to do any of that.*" [70]

**7.2 Coping strategies.** Parents described using a range of **coping strategies** [54, 61] for managing symptoms of distress and challenges during this transition. For many parents, pregnancy was seen as a time to begin healing which meant they were looking for coping strategies to support healing and change. Coping strategies included **managing, containing or controlling trauma** [45] (which links to **protecting the child and ensuring safety**, and **being a good mother**, subthemes 6.1 and 1.2), as well as specific **coping strategies during vaginal examinations** [48] and **birth** [61, 62, 64], which included consciously using **dissociation** and having **control** (subtheme 7.1 and 5.1).

"*There was one bad experience during the IVF treatments when one of the nurses rushed me. Normally I keep control of the speed of vaginal examinations and tell them to stop. One of the nurses didn't hear me, and I went into a trauma response. It's pretty spooky because the first thing that happens is, I lose eye contact, I can't communicate verbally, I just completely shut down . . . the worst extent is total paralysis. I shake uncontrollably and then I am paralyzed, I don't have any control over my whole body. My whole body is paralyzed; I can hear, but I can't talk.*" [48]

"*I really made sure, I was pretty adamant, but I was kind and I said, 'You know, doctor said I could walk, so I have to walk.' I just wanted to let everybody know I'll do anything you say, but I have to walk. I just knew if I laid in bed, I wouldn't be a very good patient, I knew that. I think now that I look back, it might have brought up memories.*" [64]

Other coping strategies parents used included **establishing healthy boundaries or balance** [61], drawing on **spirituality or faith** [67], and **self-help or care strategies** [54, 61, 67], including **reading** [53]. These strategies link to **learning about parenting** and **parenting ability** (subthemes 2.4 and 2.2).

"*My spirituality is the center of my life, and everything that I do revolves around that and my own, my beliefs, and so my world view is very much, you know, goes hand in hand with my spiritual view . . . my spiritual beliefs are what have made it possible for me to make it as far*

*as I have. You know, if I didn't believe in something I probably would have killed myself years ago, because, this world's a little bit too fucked up, life's a little bit too crazy."* [67]

*"Sometimes I just stay home and read literature . . . that kind of helps me heal."* [53]

**7.3 Factors that help recovery, growth and healing.** In line with coping strategies, parents described many factors that they felt helped recovery and healing. **Making sense of trauma** [45, 48, 54, 61, 65] was identified by many parents as critical to recovery. This links to understanding and **learning about parenting**, and the transitional role of becoming a parent (theme 2). However, parents described internal conflict between being a **'survivor' and 'victim'** [64] and there was variability in parents **understandings of trauma** [69] related to different stages of healing.

*"Knowing myself and being able to connect what I went through as a kid, and process it, and feel it, and put it out there without judgment, has completely affected who I am as a parent."* [65]

For many parents, becoming pregnant was a **catalyst to seek support** [54, 59] and many had sought **clinical therapy** [45, 59, 61, 64–66]. This links with *new beginnings* (theme 1), with some parents seeking support to repair relationships and achieve a sense of acceptance of the past and forgiveness.

*"I do want to see a counselor, because I want to make sure that I'm prepared for when the baby is born, and that I don't have any hard feelings based on my past. I want to forget about it. Forgive and forget. You can't move on. You don't move on, you're stuck and you'll always be unhappy. And then you won't be able to love yourself."* [45]

However, some parents reported **ambivalence in seeking help** despite a desire for healing [59]. There were also mixed views about the value of **peer support** [59, 72], **parenting groups** [47, 50, 53, 59] and **trauma groups** [53, 61] for helping parents make sense of trauma and developing *other relationships and support* (subtheme 3.6). Having a sense of 'normality' and an opportunity for **learning about parenting** with positive role models (which many parents described as lacking in their previous lives) were identified as important aspects of peer support and parenting groups. While having people who understood what parents had been through was an important aspect of trauma groups, there were also mixed feelings about the value of these for some parents.

*"One thing that helps me, at least for now, is that the mothers I met in the Lamaze group, we all keep in contact. We are always talking about, my son did this, or, did your son do that, and we talk about different issues of parenting. And I watch special TV programs. So, a combination of all these things are helping me to get a monitor as to what normalcy is. What I am constantly afraid of is not being able to tell normal development, even normal emotions."* [47]

*"I know it's insane, it's like how many groups do you have to talk to before you actually 'qualify', it's what I call a victim Olympics."* [61]

Parents also reported seeing value in emergent or **non-clinical therapies** [54, 59], including **art therapy** [66] and **bodywork** [54], which illustrates the challenges parents have of feeling safe while being touched [61].

"*Something that makes your body feel good and clean–just treating your body–the whole body'.*" [59]

"*But it took me a long time to be able to let somebody touch me. The first time I had a massage I was really freaking out . . . But it was really helpful. And then, it just took me a while to get used to it. And acupuncture was even scarier in some ways, and some ways not. And everything was sort of a gradation-it was all part of the healing process and I couldn't really see it at the time'.*" [54]

Other factors that parents noted as important included **allowing time** [64], **acceptance or forgiveness** [54] and **helping others** [54, 67], which was a motivation in participating in the research for some, and links to **recovery** (discussed below) and **making sense of trauma** (discussed above).

"*I've tried reading on it, but when I read on it I get depressed. I've tried therapy, I couldn't do it. So, I just try to do a little bit at a time and just not let it . . . not let it overwhelm me. Not let it dictate what I do or the decisions that I make so much.*" [64]

"*But I just made the decision that I was just going to accept it. It's a part of my life. I can't do anything about that, but I can deal with it and move on. And I basically just dealt with it in the fact that—accepting that it has happened, accepting what it has done to me, how it's changed me.*" [54]

**7.4 The healing journey though pregnancy, birth and parenting.** Many parents in this review described the experience of **healing through becoming a parent** [46, 53, 54, 59, 61, 67, 68, 73, 74] as a positive life changing journey. This links to the descriptive subtheme of *new experiences of love or joy* (subtheme 3.1), and while this brought up feelings of internal conflict and memories of their own childhood, these were largely overwhelmed by positive feelings of love for their child. This also links to *new beginnings* and *changing roles and identities* (themes 1 and 2).

"*I wouldn't be without any of them now and they're an absolute god send to me they keep me alive.*" "*It is a life changing experience you know having a baby really changes your life luckily for me it has done for the positive um. . . it was a journey a good one I wouldn't change it for the world.*" [46]

"*Now that I'm a mother, I'm a lot stronger. I am able to see rational, normal boundaries for my daughter and because of that, it's a lot easier to identify rational, sane, healthy boundaries for me. I am her mother, her protector. For a long time I thought that I was intrinsically flawed, that I came into this world somehow broken or damaged or something, and that was what drew the abuse to me. I have been sexually abused by a few people in my lifetime and irrespective of my outward appearances, of what I said or how hard I worked, there was something about me dirty and broken that I needed to hide. But when I had Emma, I realized that we all come into this world perfect, and that the flaw wasn't in me, it is in my abusers. It has been really difficult giving up this long-held belief that there was something wrong with me. I still . . . I mean there's still pieces of it that I haven't gotten rid of . . . but having Emma has allowed me to see myself differently.*" [53]

Seng [69] described this healing journey as a continuum across three stages: "1) women far along in recovery, 2) women who were not safe, and 3) women who were not ready to 'know'" (previously discussed under **avoidance** and subtheme 7.1). Parents described a healing process

of **recovery** [66, 68] and **post-traumatic growth** [68, 69]. Again, these link strongly to themes 1 and 2, with a sense of being resilient and strong.

> *"It took some years to get over and get some help and talking to support . . . now I can say I feel like I really am [over it] because I can freely speak about it, and I am not very emotional about it or depressed. I'm able to speak out and not hold or withdraw anything from anybody about it . . . Just 'cause this happened in my childhood don't mean I gotta carry it over to adulthood motherhood."* [66]

> *". . .it does have a positive impact, because I had to stop and look at myself and be like, that's not what I want to have to deal with every day. So, it made me stop and think about what I was doing with my life, and it made me change it."* [68]

Many parents emphasised the importance of **resilience or being strong** [55, 61, 66], and reporting a sense of **survival fostering resilience and agency** [47, 53]. This may relate to the need to draw on internal resources, as external support may be perceived as insufficient or create other challenges, due to relationship issues (theme 3) [61].

> *"I had to realize my own strength . . . and that's a big part of the journey . . . One of the best learning experiences I have had is becoming a mother. I have learned what I can handle, what I can deal with, how to better relate with other people through that whole state of exhaustion and you can still handle life. It's also been one of the greatest challenges that I have ever had to face which at first, yeah it's like a panic that you don't ever get away from but then as you realize what you can do, and have to do, then you become a stronger person for it as well. It was when I started having a real sense of confidence that my history started, that I started remembering things. So whether I became stronger as a person myself to feel like I could handle memories of that kind or that sort of thing."* [61]

## Summary of reflections on provider views in light of parents' views

Two studies also reported providers' views. Saewyc [67] reported views of providers for children 'ageing out' of the out of home care system. There were no major additional relevant themes or insights for this review, however the views of providers did appear a little more pessimistic than parents about the future for pregnant adolescents ageing out of care.

Garratt [51] reported views of midwives who were also child sexual abuse survivors. The midwives highlighted **barriers to being present in a professional role** and challenges when attending services for birth in **being seen as a worker and not a parent.** Midwife survivors had a unique insight into the perinatal care system, and were acutely aware of how perinatal **care can be reminiscent of abuse** and that **conventional practice is not trauma-informed**.

> *"I find it almost impossible to work on labour ward. I work on a bank contract so I can work where I want to, and I avoid delivery suite like the plague really, because I just don't want to be involved in that . . . ritualised abuse really. You know, when I think how birth can be and when I think how birth is for the majority of women now, I just don't want to be involved in that at all."* [69]

> *"A woman seeing a tube of KY gel might just freak her out. Especially if you were a child being abused and the abuser couldn't penetrate. Seeing a tube of like . . . or Vaseline, is a complete no-no."* [69]

Midwife survivors used their experiences as a positive force in guiding their approach to practice. This included giving women **choice and control**, **good communication, treating**

**women as individuals** and **acting as advocates on women's behalf**, reinforcing findings from parents in this review.

"*I think that's the biggest thing that's come out of it [being a survivor of CSA] really . . . that I want them to have some power and I want them to feel good about themselves and their body and their experiences.*" [69]

"*Then, whilst I'm actually doing it I will ask her if she wants me to talk to her while I'm doing it to tell her what I can feel or 'would you prefer me not to'? because some abusers talk through what they're doing. And that might be distressing. My father used to do that to me.*" [69]

## Discussion

### Summary of findings in response to review questions

Parents (341 mothers and 10 fathers) in 27 studies included in this review described a range of positive experiences and hopes and dreams for the future associated with becoming a parent. Parents also reported numerous challenges, at internal, interpersonal, community and societal levels. Importantly, parents identified many things they were currently doing that they felt helped them to manage distress and heal from trauma associated with child maltreatment. They also identified things that others could do to support them during the transition to parenting in pregnancy, birth and the early postpartum period.

Parents reported positive experiences of perinatal care, as well as many challenges related to fear and concerns about safety, lack of trust and control. Key factors (barriers and enablers) impacting on parents' experiences of perinatal care included: (1) *compassionate care*, or kindness, empathy and sensitivity which enabled parents to build trust and feel valued and cared for; and (2) *empowerment*: control, choice and 'having a voice', which helped parents to feel 'safe'. Parents described many mixed experiences during the transition to parenting. These included aspirations for a '*new beginning*', with pregnancy seen as an opportunity for a 'fresh start', to put the past behind them and move forward with hope for the future to create a new life for themselves and their child. Becoming a parent is also a major life transition of *changing roles and identities*, which was influenced by perceptions of the parenting role. However, parents also reported many challenges; while many knew 'what *not* to do', few had parenting role models to draw on and often lacked self-confidence about what *to* do. This anxiety was compounded by idealised notions of being a 'perfect family' and perceptions that the world was unsafe. Parents used or identified a number of important factors or strategies for '*reweaving' a future* through the complex process of managing distress and healing while becoming a parent. These included *being connected*, having positive relationships with self, baby and others; and *creating safety*, using conscious strategies to build safe places and relationships to protect themselves and their baby.

We have high confidence in the seven analytic themes that emerged from grounded theory and thematic analysis. However, parents described many personal journeys through the complex process of managing distress and healing during this major transitional period, and people's experiences were often very different. Parents often reported dealing with multiple internal conflicts which were dynamic and changing over time. There is definitely no 'one size fits all' finding in relation to the perinatal experiences of becoming a parent for those who have experienced maltreatment in their own childhoods.

### Comparisons with findings of other similar reviews

Themes emerging from this meta-synthesis are similar to those reported in a previous meta-synthesis of maternity care needs of women [37], including control, remembering,

vulnerability, dissociation, disclosure and healing. Montgomery [37] also found control and forging safe trusting relationships with care providers was important to foster a sense of safety and enable healing—as opposed to parents experiencing care as re-traumatising and reminiscent of abuse that can occur if control and trust are lacking. An important consideration is that we found a very limited number of studies involving fathers and other family members, which is similar to what has been reported in another review [35] and study [76]. We note that our themes align with some of the findings of male parenting experiences, for example being influenced by self-perceptions as adequate parents, conceptualisation of parenting as a potential healing experience and source of strength, fear of repeating the past, and relationship challenges. However, it is clear that there is a great need to develop deeper understandings of the parenting experiences of fathers who have experienced childhood maltreatment and may be suffering from the impacts of complex trauma. There were also no included studies involving parents with sexual diversity. However, a recent study of queer-identifying mothers highlights many of the issues identified in this review as important, with these women experiencing greater marginalisation within the maternity care system [77]. Olsen [78] found limited evidence of screening for child maltreatment during routine prenatal care, but that qualitative evidence suggested women were likely to be supportive of screening in prenatal care. Our findings identified a number of barriers to disclosure of abuse and we suggest these concerns should be considered to foster safety and minimise risks associated with routine prenatal screening for trauma.

In this review, we excluded studies that included parents where there is likely to be a high prevalence of child maltreatment and/or complex trauma but which was not explicitly identified (e.g. parents experiencing substance use, homelessness, incarceration, or categorised 'at risk' with regards to their parenting). We note that there is overlap in the findings of studies and reviews in these population groups and this current review, and some of these studies are listed in the table of excluded studies (see S7 Appendix).

## Reflections on implications of findings in relation to key frameworks and theories

The findings in this review reinforce the importance of the transition to parenting as a critical life course opportunity for supporting parents who have experienced child maltreatment in their own childhood. 'Life course approaches' involve the study of long-term impacts of early life exposures to identify critical periods of vulnerability and understand pathways to inequities in later life, so that effective strategies for prevention can be understood. It combines epidemiological [79] and social policy research [80], and increasingly incorporates socio-ecological models to describe multifactorial and intergenerational influences and processes [81]. The perinatal period is a critical life course point for parents who have experienced child maltreatment for several reasons. Parents are filled with hope for a 'new beginning' and are strongly motivated to provide a better life for their child, but they also face numerous challenges, including managing trauma symptoms that may be triggered during perinatal care and/or bodily experiences associated with pregnancy, birth and breastfeeding. Any effective support strategies are highly likely to have a direct and ongoing impact on the parent's future social, emotional and physical wellbeing. Becoming a parent is also the first time since childhood (when the maltreatment occurred) that most people have regular health service contacts, offering practical opportunities to provide and facilitate access to appropriate support. Thus, perinatal care providers should be aiming to demonstrate leadership in fostering safety within the health and social care system, and providing 'trauma-informed care' to prevent re-traumatisation and compounding of complex trauma related distress. Parents in this review have

highlighted the need for both compassionate care and practical support. The perinatal period is also the most effective for promoting wellbeing and prevention of intergenerational trauma being transmitted to the infant. There is strong international agreement that any early investments to promote nurturing care are returned many fold in terms of future social, emotional and physical wellbeing [82–84].

As outlined above, life course approaches increasingly incorporate socioecological models, and these are particularly important in framing and understanding the context of child maltreatment, complex trauma and the sequelae across the life course. The WHO use these models to explain why violence, including family violence and child maltreatment, are more likely to occur in certain contexts [1]. Addressing these contexts is critical to addressing violence. Parents in this review described in detail a range of challenges across all socioecological levels. These include managing *internal* conflicts associated with distress and healing that often draws or relies on internal resources due to limited access to *community*-level factors and resources. Parents also described challenges negotiating *interpersonal* relationships. These factors impact on the transitioning role experience and a sense of safety, which are compounded by *societal* factors including stigma and discrimination, education and employment opportunities, and housing. Thus, planning models that facilitate consideration of socio-ecological factors, such as Intervention Mapping [85], are critical. Findings in this review also reinforce the Australian National Complex Trauma guidelines principles [10] of *safety* (physical and emotional), *trustworthiness*, *choice*, *collaboration*, and *empowerment*, including sharing power, ensuring parent control and skill building–particularly in relation to parenting skills.

We have less confidence in the relevance of the findings of this review for ethnic minority and Indigenous parents as there is limited representation within included studies in this review. Despite this, we note that our findings support recently developed values that were incorporated into a Conceptual Framework for co-designing perinatal awareness, recognition, assessment and support strategies for Aboriginal parents in Australia [16]. These include *compassion*, *culture* (and spirituality) and a *holistic* approach to care which is consistent with Aboriginal understandings of social and emotional wellbeing [86]. Such understandings frequently highlight connectedness as a central tenet–where mental health, physical health, connection to family, community, culture, spirituality/ancestors and land are integral to wellbeing and heavily influenced by broader cultural, political, historical and social determinants [87]. The findings in this review that included predominantly non-Indigenous parents raises at least two important questions with regards to informing the Healing the Past by Nurturing the Future project [16]: (1) How relevant are our findings to Indigenous parents with histories of trauma?; and (2) How relevant might Indigenous models of wellbeing be for understanding and conceptualising healing from complex trauma in other populations?

Parents descriptions of symptoms of distress in this review appear consistent with the domains of distress recently included in the new WHO *ICD 11* criteria for complex trauma. This includes existing criteria for PTSD (intrusions or 're-experiencing the events (triggers), avoidance, a 'sense of threat' impacting on mood, cognitions, arousal and reactivity); and additional criteria for complex trauma of 'affect/emotional dysregulation', 'negative self-concept' and 'relational disturbances' [7]. Additional domains of trauma-related distress have also been identified in Aboriginal populations, who have experienced historical trauma and violence, state-sanctioned removal of children from families, and systematic attempts to destroy culture [17]. These additional areas of distress include: feeling isolated or disconnected from community; identity loss or fragmentation/confusion; unresolved grief and loss; substance abuse, self harm or violence against others; and suicidality [20].

While complex trauma related distress and 'diagnostic criteria' have potential value in helping to identify people who may benefit from support, we also note the current debates about

the use of diagnostic approaches in conceptualising and addressing mental health. The British Psychological Association have proposed the Power Threat Meaning Framework (PTMF) [88]. The PTMF is "an over-arching structure for identifying patterns in emotional distress, unusual experiences and troubling behaviour, as an alternative to psychiatric diagnosis and classification" [88, p 5], which contrasts with current international debates regarding classification of complex trauma. It reframes behaviours related to complex trauma as natural self-protective responses to a threat rather than a pathological deficit. Essentially asking the question of "What has happened to you?", rather than 'What is wrong with you?" [88]. The framework is based on four inter-related aspects: the operation of Power; the Threat the operation of power poses; the central role of Meaning (shaped by social and other discourses); and the learned and evolved Threat Responses to these inter-related elements [88]. This framework provides a constructive and positive approach for conceptualising the experiences and views expressed by parents in this review–for example, thinking about how Power has operated in parents' lives and current responses to authority during perinatal care and the transition to becoming a parent; thinking about how that Power is operating now; how parents experiences have shaped Meaning; and that the Threat Responses they are experiencing are natural responses to this situation, grounded in a sense of fear and feeling unsafe [88]. Importantly, this strengths-based approach may help parents to feel they, and the responses and feelings of distress they are experiencing, are normal. Aspirations for 'being normal' were particularly important for parents in this review as they worked through this important transitional period of becoming a parent with evolving roles and identities.

While there are discussions about introducing routine prenatal screening for child maltreatment experiences [78], it is particularly important that the principles of population-based screening are carefully considered in this context [89, 90]. This involves comprehensive assessment of both the benefits and risks, including epidemiology, current screening practice, acceptability, efficacy and cost, and—critically–the availability of acceptable and effective support for parents. The importance of availability of support as a prerequisite for screening was highlighted in an included study in this review [69, 91]. Findings in this review also highlighted many concerns about disclosure (including delayed memory), raising questions about the potential accuracy of any routine screening approach. This reinforces suggestions by others [48] to adopt 'universal precautions' with provision of sensitive trauma-informed perinatal care for all women. In Australia, Atkinson et al [92, 93] propose an 'educaring approach' that combines compassionate care and deep listening with gentle 'education', to help people understand and make meaning of their trauma experiences, and find ways to heal or 'reweave' a future. This analogy of 'reweaving' was developed by Pitre [75] and refers to a process of reconstructing self and a world that is safe while mothering with a background of childhood violence. Pitre [75] argues that many clinical routines and practices appear to meet professional need rather than provide the individualised support that is required and reinforce rigid and unhelpful misconceptions of what it means to be a 'good' mother. These misperceptions were a serious challenge for many parents in this review, and are also likely to impact on the accuracy of screening disclosure. Concerns about screening are particularly important in the context of Aboriginal health, and Scrimgeour (1996) argues that additional criteria should be applied for screening programs among Aboriginal people, ensuring that the benefits outweigh the risks and additional surveillance on their lives.

## Strengths and limitations

A major strength of this review is that it was developed following a comprehensive scoping review [6], which helped to gain familiarity with the literature, synthesis challenges, and refine

a comprehensive search strategy and identify a large volume of studies. There is always the potential for reporting bias and social desirability bias in research on sensitive topics like those included in this review. Given the lack of information regarding the relationship between researcher and participants in 13 studies, this was a particular concern in this synthesis. However, it is clear from the richness of the evidence presented within the papers and frankness and openness of the participants, that reporting bias or social desirability bias did not appear to overly affect the studies. It is possible some studies may have been missed, however the authors feel we reached sufficient 'saturation' to attain a high level of confidence in the analytic themes. However, most studies were drawn from specialised clinical settings and predominantly involved mothers who had experienced child sexual abuse. Additionally, the search was restricted to studies published in English for pragmatic reasons. It is possible there may be different themes from non-English studies owing to the different cultural contexts they may be experiencing. Hence we are less confident in the relevance of the findings for fathers, parents from ethnic minority or Indigenous populations, young parents, single parents, parents who speak a language other than English and parents who have experienced types of child maltreatment other than child sexual abuse.

## Implications for practice, policy and research

Practice implications of these findings include the need to consider four main domains in perinatal care in relationship to child maltreatment: *awareness* (to minimise the risk of retraumatisation), safe *recognition* processes to minimise the risk of harm from disclosure, appropriate and effective *assessment* approaches, and a broad range of support strategies that include but are not limited to clinical therapies [16]. Our findings reinforce Muzik et al's [59] and Seng's [69] guidance for trauma-informed perinatal care and the need for 'hope affirming practices' that nurture the hopes and dreams of 'new beginnings' for parents who have experienced maltreatment in their own childhood. Findings in this review also point to 'continuity of care' models of perinatal care that have been highly acceptable and effective in improving perinatal care experiences and outcomes [94, 95], and are likely to be particularly important for parents who have experienced child maltreatment and are experiencing complex trauma. Continuity of care models can help to enable the development of trusting relationships with providers and foster a sense of safety in perinatal care [96]. Other potentially relevant strategies that appear to be effective in perinatal care for improving infant outcomes generally include: home visiting; antenatal and postnatal education and/or support; massage; support to prevent postnatal depression; treating maternal depression in the perinatal period; neonatal behavioural assessment; enhancing sensitivity and/or attachment security; and support for low income/socially disadvantaged parents [97]. Fragmentation of care is also particularly relevant for parents with complex needs, and strategies to actively improve coordination and collaboration–such as holistic care models—are urgently needed [98].

Policy implications of this review include the need for broad social policies that help to reassure parents that 'the world can be a safe place' for them and their children. Concerns about child protection were highlighted by parents in this review, which highlights the strengths of early investment in 'normalising' support which fosters a sense of safety, within 'mainstream' care systems. The importance of investment in broader societal factors (housing, education, safety, addressing stigma) were also highlighted in this review, and these investments are likely to have important returns in terms of both secondary prevention for the parent and primary prevention for the infant [99].

Research implications include an urgent need to build evidence of effective strategies across four key domains of perinatal awareness, recognition, assessment and support. This includes

conducting research that is strongly grounded in what parents actually want and are currently doing, and not restricting research solely to clinical therapies. It is important to recognise that while this is a relatively new area for international researchers, policy makers and clinicians, many parents have been finding ways to effectively navigate the challenges outlined in this review for a long time. There are many challenges to implementation and evaluation of complex interventions. Given the current limited evidence of safety or effectiveness of 'interventions' in this area, incorporating the use of flexible evaluation plans with short reflective cycles (e.g. action research) may help to identify any unforeseen problems early and respond quickly to minimise risks of harm. We also note the paucity of studies and the urgent need to expand the applicability of the evidence-base in this area of research, including drawing on community-based samples that include fathers and other family members, ethnic minority and Indigenous populations, single parents, young parents and parents with types of maltreatment other than child sexual abuse. The findings from this review will be used to inform a subsequent systematic review of perinatal interventions to support parents who have experienced maltreatment in their own childhoods. The review will include a search for interventions based on what parents have said they want and currently do in this review.

## Conclusions

The perinatal period and transition to parenting is a unique life course opportunity to provide support for parents with a history of maltreatment in their own childhoods, and to promote relational healing and prevent intergenerational transmission of trauma. Developing a better understanding of parents' experiences and views of perinatal care and early parenting, and the strategies that parents use to ensure a safe and nurturing environment for their children, is critical for informing the development of acceptable and effective support strategies. This review provides a comprehensive overview of these experiences and views, and highlights the need for compassionate, empowering and safe perinatal support through this transitional period to create a 'fresh start' and 'reweave a future' for themselves and their families.

## Supporting information

**S1 Appendix. PRISMA checklist.** The PRISMA checklist completed for this review.
(DOC)

**S2 Appendix. Sample search strategy–PsychInfo.** A sample of the search strategy used by this review with the PsychInfo database.
(DOCX)

**S3 Appendix. NVivo file attributes.** A copy of the Nvivo file attributes imported from the Excel data extraction spreadsheet.
(DOCX)

**S4 Appendix. Concept map.** The concept map of analytic themes and descriptive subthemes generated by this review.
(JPG)

**S5 Appendix. Summary analytic themes, subthemes, and axial codes.** Summary table of analytic themes, descriptive subthemes, and axial codes generated by this review.
(DOCX)

**S6 Appendix. Summary analytic themes and descriptive subthemes by selected study characteristics.** Summary table considering review's analytic themes and descriptive subthemes by

selected study characteristics.
(DOCX)

**S7 Appendix. Table of excluded studies.** Table of excluded studies, with reasons for exclusion provided.
(DOCX)

**S8 Appendix. Draft findings with overinclusive quotes.** Provides a long version of draft qualitative findings with overinclusive list of supporting parent and study author quotes.
(DOCX)

## Acknowledgments

We thank Dr Sharinne Crawford, Ms Steph Saidel and Ms Karen Glover for their assistance screening titles and abstracts for inclusion in this review. We gratefully acknowledge the support and advice of Mr Stav Amichai Hillel for assistance running some of the searches and Ms Jenny Fafeita with using NVivo for this review. We thank Dr Fiona Mensah for comprehensive feedback on the final draft of this review. We acknowledge the oversight of this review from other members of the Healing the Past by Nurturing the Future Investigator Group who are not listed as authors: Judy Atkinson, Jan Nicholson, Deirdre Gartland, Helen Herrman, Karen Glover, Tanja Hirvoven, Fiona Mensah, Caroline Atkinson, Shawana Andrews, Helen McLachlan, Sandra Campbell and Danielle Dyall. We also acknowledge the representatives of our project partner organisations for their project oversight and leadership in this area, including: Jesse Odgers (Victorian Aboriginal Community Controlled Health Organisation), Gina Bundle (Royal Womens Hospital), Sarah Crossing and Joy Makepeace (Nunkuwarrin Yunti of South Australia Inc.), Jacqui Ah Kit (Women and Children's Health Network), Bronwyn Silver (Central Australian Aboriginal Congress) and Alison Elliott (Bouverie Family Healing Centre). We thank Rhonda Marriott for her 'critical friendship' and other team members including Carol Reid, Georgie Igoe and Leanne Slade for input into team discussions related to this review.

## Author Contributions

**Conceptualization:** Catherine Chamberlain, Claire Stansfield, Katy Sutcliffe, Stephanie J. Brown, Sue Brennan.

**Data curation:** Catherine Chamberlain, Naomi Ralph, Stacey Hokke, Graham Gee, Claire Stansfield, Stephanie J. Brown, Sue Brennan.

**Formal analysis:** Catherine Chamberlain, Naomi Ralph, Stacey Hokke, Graham Gee, Katy Sutcliffe, Stephanie J. Brown, Sue Brennan.

**Funding acquisition:** Catherine Chamberlain, Stephanie J. Brown.

**Investigation:** Catherine Chamberlain.

**Methodology:** Catherine Chamberlain, Katy Sutcliffe, Stephanie J. Brown, Sue Brennan.

**Project administration:** Catherine Chamberlain, Naomi Ralph, Stacey Hokke, Yvonne Clark, Graham Gee, Sue Brennan.

**Resources:** Catherine Chamberlain.

**Software:** Catherine Chamberlain.

**Supervision:** Catherine Chamberlain, Katy Sutcliffe, Stephanie J. Brown, Sue Brennan.

**Validation:** Catherine Chamberlain.

**Visualization:** Catherine Chamberlain, Stacey Hokke.

**Writing – original draft:** Catherine Chamberlain, Stacey Hokke, Claire Stansfield.

**Writing – review & editing:** Catherine Chamberlain, Naomi Ralph, Stacey Hokke, Yvonne Clark, Graham Gee, Claire Stansfield, Katy Sutcliffe, Stephanie J. Brown, Sue Brennan.

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
