## [Decision Letter · Decision Letter 0]

10 Oct 2019

PONE-D-19-20272

Healing The Past By Nurturing The Future: a qualitative systematic review and meta-synthesis of pregnancy, birth and early postpartum experiences and views of parents with a history of childhood maltreatment

PLOS ONE

Dear A/Prof Chamberlain,

Thank you for submitting your manuscript to PLOS ONE. After careful consideration, we feel that it has merit but does not fully meet PLOS ONE’s publication criteria as it currently stands. Therefore, we invite you to submit a revised version of the manuscript that addresses the points raised during the review process, especially to Reviewer 2.

We would appreciate receiving your revised manuscript by Nov 24 2019 11:59PM. To enhance the reproducibility of your results, we recommend that if applicable you deposit your laboratory protocols in protocols.io, where a protocol can be assigned its own identifier (DOI) such that it can be cited independently in the future. For instructions see: http://journals.plos.org/plosone/s/submission-guidelines#loc-laboratory-protocols

We look forward to receiving your revised manuscript.

Kind regards,

Ju Lee Oei

Academic Editor

PLOS ONE

Journal Requirements:

2. We noticed you have some minor occurrence of overlapping text with previous publications, which needs to be addressed. In your revision ensure you cite all your sources (including your own works), and quote or rephrase any duplicated text outside the methods section.

4. Please ensure that you refer to Figure 1 in your text as, if accepted, production will need this reference to link the reader to the figure.

Additional Editor Comments (if provided):

Reviewers' comments:

Reviewer's Responses to Questions

**Comments to the Author**

1. Is the manuscript technically sound, and do the data support the conclusions?

Reviewer #1: Yes

Reviewer #2: Yes

2. Has the statistical analysis been performed appropriately and rigorously? 

Reviewer #1: N/A

Reviewer #2: N/A

3. Have the authors made all data underlying the findings in their manuscript fully available?

Reviewer #1: Yes

Reviewer #2: Yes

4. Is the manuscript presented in an intelligible fashion and written in standard English?

Reviewer #1: Yes

Reviewer #2: Yes

5. Review Comments to the Author

Reviewer #1: Thank you for the opportunity to review this paper. Overall this paper demonstrates a sound command of the English language and offers new insights that contribute to extant literature surrounding the pervasive effects of developmental trauma.

Background: the context and topic are clearly presented. Line 196, Understand should read understanding

Methods: clear description of the search strategies and eligibility criteria. It is unclear how an adoptive or foster parent could be experiencing pregnancy and birth. What was the rationale for excluding case studies? Good description of the analytic methods used. Strategies to ensure the rigour of this review are well explained.

Findings: well presented, clearly articulated and important.

This paper makes a significant contribution to the ongoing discussions of developmental trauma. It provides new insights into the adult experience after experiencing developmental trauma. These insights are important for both health and social services providers.

Well done – it was a pleasure to read this paper. I look forward to the next review.

Reviewer #2: This review aimed to understand the pregnancy, birth and early postpartum experiences of patients who had reported experiencing maltreatment in childhood. The authors have done a commendable job of addressing this by assembling a tremendous amount of information concerning a complex array of factors and synthesizing a comprehensive document that provides a valuable resource of information. The data has been well presented and, although the applicability of the findings are limited by the paucity of studies in this field, this review helps to identify key areas where future research should be directed. Inclusion of a few clarifications and minor revisions would make this paper suitable for publication.

Major comments:

1) In lines 439-447, the authors discuss the types of abuse reported in the studies included; however, it would be helpful to briefly touch upon the classification of ‘child maltreatment’ and the types of abuse it includes in the introduction as it may be perceived as a fairly broad term including a range of behaviours. It would also set the premise for readers unfamiliar with this field of research.

2) Although the reasons for excluding individual studies have been summarised in the supplementary files and have also been briefly mentioned in the discussion, were there any exclusion criteria that had been identified prior to development of the search strategy? If yes, it would be useful to include this information in the methods section.

3) In line 291, the authors acknowledge that their search strategy was limited to studies reported in English only. I was wondering if they had any estimates of the number of papers in other languages they could have potentially ‘missed’ and how this may have affected their findings? This should also be acknowledged as a limitation in the discussion.

4) It would be useful to include the date the search strategy was last run. Were any attempts made to contact the authors of the included studies to identify further studies?

5) Although the risk of bias was assessed, it would also be worthwhile to reflect upon potential sources of bias at the individual study level (and include this in Table 1 where possible) and consider how these may have affected the findings of the meta-synthesis. For example, the majority of studies included seemed to have collected information via open or semi-structured individual interviews, leading one to question the possibility of reporting bias given the sensitive nature of the issues being addressed. The authors have also not discussed potential sources of bias in the review itself (e.g those introduced by the language limitation of the search strategy), and inclusion of this in the discussion is recommended.

Minor comments

1) There are various typos and minor errors throughout the document here (e.g line numbers 146, 196, 313, 437 to list a few).

2) The line numbering appears to end at 506, with the remainder of the document not being numbered. Please modify this to ensure consistency.

6. PLOS authors have the option to publish the peer review history of their article (what does this mean?). If published, this will include your full peer review and any attached files.

Reviewer #1: No

Reviewer #2: No

---

## [Author Response · Author response to Decision Letter 0]

31 Oct 2019

Response to Reviewers

Journal Requirements

We have ensured that the manuscript meets PLOS One style requirements, including those for file naming.

2. We noticed you have some minor occurrence of overlapping text with previous publications, which needs to be addressed. In your revision ensure you cite all your sources (including your own works), and quote or rephrase any duplicated text outside the methods section.

We thank the editor for noting the minor occurrence of overlapping text, and we have processed the manuscript in ‘ithenticate’ and rephrased and/or added references throughout the manuscript accordingly. 

The minor overlapping text occurrences were predominantly with the previous scoping review which had informed the development of the methods for this in-depth qualitative meta-synthesis. We are happy to discuss any further edits required with the editorial team. 

3. Please include captions for your Supporting Information files at the end of your manuscript, and update any in-text citations to match accordingly.

Captions have been provided for each of the Supporting Information files at the end of the manuscript, and all in-text citations have been matched accordingly. 

4. Please ensure that you refer to Figure 1 in your text as, if accepted, production will need this reference to link the reader to the figure.

Figure 1 is now referred to in text, at line 383 (Track changes version).

Reviewer #1

Thank you for the opportunity to review this paper. Overall this paper demonstrates a sound command of the English language and offers new insights that contribute to extant literature surrounding the pervasive effects of developmental trauma.

We thank reviewer #1 for these positive comments. 

5. Background: the context and topic are clearly presented. Line 196, Understand should read understanding

We have made this edit at line 196 (164 in track changes version), changing ‘Understand’ to ‘Understanding’.

6. Methods: clear description of the search strategies and eligibility criteria. It is unclear how an adoptive or foster parent could be experiencing pregnancy and birth. 

Yes we agree with reviewer #1 that this is a bit unclear for this initial review and is likely to be more relevant for the next review (birth to two years postpartum). However, we have used the same inclusion criteria and it was not possible to include studies which exclusively only included parents during pregnancy, birth or breastfeeding (due to participants within included studies. Hence the rationale for noting (line 218 track changes version): “This review includes studies exclusively or predominantly including parents during pregnancy, birth and early postpartum period (up to approximately six weeks after birth); or where the focus of the study was specifically on pregnancy, birth or breastfeeding.” Hence we have used an overinclusive approach to account for unlikely but possible studies. 

7. What was the rationale for excluding case studies? 

There are several reasons why we decided to exclude ‘case studies’ in this review. First, in considering case studies in the scoping phase, we felt that case studies in this field tend to have more of a focus on ‘clinical observations’ and views of clinicians, and we explicitly wanted the focus of this review is on the views and experiences of parents rather than clinicians. Additionally, we had such a large volume of qualitative data from the qualitative studies conducted with parents (hence had to ‘split’ the original review into two parts) that we felt confident we would comfortably reach saturation of the thematic categories without including case studies. 

We have added this rationale (line 253 track changes version): 

“Single case studies involving one or two parents were excluded, as these tended to have more of a focus on clinician observations, and the aim of this review is to understand parent views and experiences, and there was already sufficient data for saturation of thematic categories.” 

Good description of the analytic methods used. Strategies to ensure the rigour of this review are well explained. 

Findings: well presented, clearly articulated and important.

This paper makes a significant contribution to the ongoing discussions of developmental trauma. It provides new insights into the adult experience after experiencing developmental trauma. These insights are important for both health and social services providers.

Well done – it was a pleasure to read this paper. I look forward to the next review.

We thank reviewer #1 for the constructive comments and positive feedback. 

Reviewer #2

This review aimed to understand the pregnancy, birth and early postpartum experiences of patients who had reported experiencing maltreatment in childhood. The authors have done a commendable job of addressing this by assembling a tremendous amount of information concerning a complex array of factors and synthesizing a comprehensive document that provides a valuable resource of information. The data has been well presented and, although the applicability of the findings are limited by the paucity of studies in this field, this review helps to identify key areas where future research should be directed. Inclusion of a few clarifications and minor revisions would make this paper suitable for publication.

We thank reviewer #2 for this feedback. 

Major comments:

8. In lines 439-447, the authors discuss the types of abuse reported in the studies included; however, it would be helpful to briefly touch upon the classification of ‘child maltreatment’ and the types of abuse it includes in the introduction as it may be perceived as a fairly broad term including a range of behaviours. It would also set the premise for readers unfamiliar with this field of research.

We have added the following definition of child maltreatment to the introduction (see line 110 track changes version), to provide sufficient detail for readers unfamiliar with the field of research, as follows:

“In this review we use the term and focus on parents that have experienced ‘child maltreatment’ to reflect the existing literature, recognising the current dynamic state of definitions and terms being proposed. We use a definition of child maltreatment consistent with the World Health Organization (WHO) to include “all types of physical and/or emotional abuse and neglect, and sexual abuse that results in actual or potential harm of children in the context of relationships of responsibility, trust and power” [1].

9. Although the reasons for excluding individual studies have been summarised in the supplementary files and have also been briefly mentioned in the discussion, were there any exclusion criteria that had been identified prior to development of the search strategy? If yes, it would be useful to include this information in the methods section.

We did not apply any exclusion criteria prior to the development of search strategy for this review. The search strategy was developed with the purpose of identifying potentially relevant studies for inclusion only. Hence we have not included any additional information in relation to this query.

10. In line 291, the authors acknowledge that their search strategy was limited to studies reported in English only. I was wondering if they had any estimates of the number of papers in other languages they could have potentially ‘missed’ and how this may have affected their findings? This should also be acknowledged as a limitation in the discussion.

Qualitative meta-synthesis is intended to capture an unbiased sample of people's views, with the emphasis on the obtaining ‘saturation’ on a range of themes, rather than on detecting comprehensiveness of all studies that exist as is required for meta-analysis of efficacy studies. 

However, our searches in English language databases were highly sensitive and aimed to capture a range of populations and themes within the scope of the review. Any ‘missed’ studies that would affect the findings would need to provide different themes rather than provide more of the same studies. It is impossible to estimate what themes we do not know about, but the reduction of studies in the preliminary search from applying this limitation was very small (i.e. <600 articles from >8,000 in the sample Psychinfo search). It is possible there may be different themes from non-English language studies owing to the different cultural contexts they may be describing. However, this is often a feature of qualitative synthesis.

We have acknowledged this as a limitation in the discussion (see line 1691 track changes version), as follows:

“Additionally, the search was restricted to studies published in English for pragmatic reasons. It is possible there may be different themes from non-English studies owing to the different cultural contexts they may be experiencing. Hence we are less confident in the relevance of the findings for fathers, parents from ethnic minority or Indigenous populations, young parents, single parents, parents who speak a language other than English and parents who have experienced types of child maltreatment other than child sexual abuse.” 

11. It would be useful to include the date the search strategy was last run. 

The date that the search strategy was last conducted (22/6/2018), was noted in the original manuscript submitted at line 295, and appears now in the revised manuscript at line 266 (track changes version” as follows:. 

“We searched for potentially relevant studies from databases and other sources from the date of database inception up to 22 June 2018.”

12. Were any attempts made to contact the authors of the included studies to identify further studies?

No we did not contact authors of included studies to identify further studies. As previously noted in response to comment 10, we conducted a highly sensitive search and established an autoalert to identify newly published studies as they become available. We did however, search the reference lists of relevant reviews to identify any studies that may have been missed, and were reassured about our search when no new studies were identified from these sources. 

13. Although the risk of bias was assessed, it would also be worthwhile to reflect upon potential sources of bias at the individual study level (and include this in Table 1 where possible) and consider how these may have affected the findings of the meta-synthesis. For example, the majority of studies included seemed to have collected information via open or semi-structured individual interviews, leading one to question the possibility of reporting bias given the sensitive nature of the issues being addressed. 

We thank the reviewer for this comment and opportunity to describe in more detail how we have carefully considered potential sources of bias both within and across studies for this review. We agree that these individual study assessments are important. 

Rather than add more detail to Table 1, we have added the references to the following description in line 443-453: 

“Consideration of the relationship between the researcher and participants (item 6) was the exception (not discussed in eight studies [52, 54, 61, 64, 65, 68, 73, 75] and unclear in five studies [50, 59, 62, 70, 76]). There is always a concern with research that participants may feel reluctant or unable to be frank or open about the challenges they face or they might provide socially desirable responses because of the sensitive nature of the topic. However, this was not rated as serious concern in many of these studies due to the richness and frankness of data presented. Of the 27 studies, 15 were rated as of high methodological quality overall (i.e. no or very minor concerns for any of the CASP items), 10 as moderate (due to moderate to serious concerns about consideration of research-participant relationship [52, 61, 64, 68], adequacy of data [56, 58], unclear sampling [71, 74, 75], or synthesis/analysis of data [53]), one as low and one as very low (Table 1).

We have also added the following text to the discussion to consider the impact of methodological limitation (sources of bias) on the findings of the meta-synthesis (1691):

“There is always the potential for reporting bias and social desirability bias in research on sensitive topics like those included in this review. Given the lack of information regarding the relationship between researcher and participants in 13 studies, this was a particular concern in this synthesis. However, it is clear from the richness of the evidence presented within the papers and frankness and openness of the participants, that reporting bias or social desirability bias did not appear to overly affect the studies.”. We have also indicated that the study level assessments are available on request. The rationale for our approach to considering and reporting study level assessments is as follows.

The Critical Appraisal Skills Programme (CASP) checklist that we used to assess methodological quality of studies prompts reviewers to consider the biases mentioned (e.g. under domains relating to the research-participant relationship and the method of data collection). Since most studies were of high methodological quality (i.e. no important concerns were identified), and the assessments were formally integrated into our rating of overall confidence in each finding (the GRADE Cerqual assessment), we felt there was limited additional value in reporting the detailed assessment for individual studies. The GRADE Cerqual process considers how these study limitations may have affected the findings of the meta-synthesis in Table 2, for each of the analytic themes (see description of how this was conducted on line 355 track changes version). Since the GRADE Cerqual assessment relates directly to each meta-synthesis finding, and considers the richness of data (data adequacy - which reporting bias would affect), coherence and relevance, in addition to the methodological limitations (column 3), it provides the most direct information about the confidence that can be placed in each finding. Our assessment included ‘sensitivity analysis’ of the impact of removing studies with methodological concerns (mainly related to lack of reflexivity). However, due to the large volume of data and saturation of the analytic themes the methodological limitations in some studies did not affect our confidence in the findings across studies.

14. The authors have also not discussed potential sources of bias in the review itself (e.g those introduced by the language limitation of the search strategy), and inclusion of this in the discussion is recommended.

Please see response to comment 10 above. 

Minor comments

15. There are various typos and minor errors throughout the document here (e.g line numbers 146, 196, 313, 437 to list a few).

We thank the reviewer for noting some typos and minor errors and have corrected them in text. Three authors have also reviewed the document carefully and we hope that there are no remaining errors, or that any remaining typos are minor and can be identified in the copy-editting phase if accepted. 

16. The line numbering appears to end at 506, with the remainder of the document not being numbered. Please modify this to ensure consistency.

The line numbering has been modified for consistency.

---

## [Editor Report · Decision Letter 1]

6 Nov 2019

Healing The Past By Nurturing The Future: a qualitative systematic review and meta-synthesis of pregnancy, birth and early postpartum experiences and views of parents with a history of childhood maltreatment

PONE-D-19-20272R1

Dear Dr. Chamberlain,

We are pleased to inform you that your manuscript has been judged scientifically suitable for publication and will be formally accepted for publication once it complies with all outstanding technical requirements.

With kind regards,

Ju Lee Oei

Academic Editor

PLOS ONE
---

## [Editor Report · Acceptance letter]

6 Dec 2019

PONE-D-19-20272R1 

Healing The Past By Nurturing The Future: a qualitative systematic review and meta-synthesis of pregnancy, birth and early postpartum experiences and views of parents with a history of childhood maltreatment 

Dear Dr. Chamberlain:

I am pleased to inform you that your manuscript has been deemed suitable for publication in PLOS ONE. Congratulations! Your manuscript is now with our production department. 

With kind regards,

on behalf of

Dr. Ju Lee Oei 

Academic Editor

PLOS ONE